# B cell class switch recombination is regulated by DYRK1A through MSH6 phosphorylation

Liat Stoler-Barak[1], Ethan Harris[2], Ayelet Peres ®[3], Hadas Hezroni[1], Mirela Kuka[4], Pietro Di Lucia[4], Amalie Grenov ®[1], Neta Gurwicz[1], Meital Kupervaser[5], Bon Ham Yip[2], Matteo Iannacone ®[4,6], Gur Yaari[3], John D. Crispino ®[2] & Ziv Shulman ®[1] ✉

Protection from viral infections depends on immunoglobulin isotype switching, which endows antibodies with effector functions. Here, we find that the protein kinase DYRK1A is essential for B cell-mediated protection from viral infection and effective vaccination through regulation of class switch recombination (CSR). *Dyrk1a*-deficient B cells are impaired in CSR activity in vivo and in vitro. Phosphoproteomic screens and kinase-activity assays identify MSH6, a DNA mismatch repair protein, as a direct substrate for DYRK1A, and deletion of a single phosphorylation site impaired CSR. After CSR and germinal center (GC) seeding, DYRK1A is required for attenuation of B cell proliferation. These findings demonstrate DYRK1A-mediated biological mechanisms of B cell immune responses that may be used for therapeutic manipulation in antibody-mediated autoimmunity.

Effective long-lasting protection from invading pathogens largely depends on the generation of antibodies by the infected host[1,2]. In addition, antibodies can have a function in the clearance of invading microbes in a primary immune response[3] through pathogen neutralization activity, and induction of a range of cell-mediated effector functions[4–6]. These include NK-mediated killing of infected cells, and pathogen clearance by phagocytes through the interaction of the Fc part of the immunoglobulin with Fc-receptors that are expressed on immune cells[4,5,7]. Furthermore, the Fc region of the antibody can activate the complement system, which involves a series of enzyme-mediated cleavage activities that can lead to the killing of the target cells[8]. Similar antibody functions have a role in the clearance of aberrant self cells, such as malignant tumors,[6] and induce tissue damage in autoimmune diseases[5].

Antibody effector functions are determined by their isotype class. Prior to antigen encounters, naive B cells express both IgM and IgD B-cell receptors (BCRs) on their surface[9,10]. Following cognate antigen interaction, and in response to mitogens and specific cytokines, B cells can switch their immunoglobulin isotype through a process known as class switch recombination (CSR)[11,12]. This mechanism involves the generation of nucleotide mismatches at the immunoglobulin switch regions by deamination of cytidine to uracil through activation-induced cytidine deaminase (AID) activity, generation of DNA breaks, and activation of DNA repair mechanisms[11,13–17].

Germinal centers are the major source of class-switched and long-lived plasma cells. These are microanatomical sites that are seeded by antigen-specific B cells about 5 days after pathogen infection or vaccination[9]. In these niches, B cells mutate their immunoglobulin genes followed by B-cell receptor (BCR)-affinity-based selection for clonal expansion and differentiation into plasma cells (PCs)[18]. Within the GC, T follicular helper cells select B cells for clonal expansion through triggering CD40 and ICOSL activation on GC B cells and through cytokine secretion. These signals increase the cell division rate of the selected clones through the initial triggering of Myc transcription and downstream genetic programs within the GC light zone

[1]Department of Systems Immunology, Weizmann Institute of Science, Rehovot 7610001, Israel. [2]Department of Hematology, St. Jude Children's Research Hospital, Memphis, TN 38105, USA. [3]Faculty of Engineering, Bar Ilan University, Ramat Gan 52900, Israel. [4]Vita-Salute San Raffaele University and Division of Immunology, Transplantation and Infectious Diseases, IRCCS San Raffaele Scientific Institute, Milan, Italy. [5]De Botton Institute for Proteomics, Grand Israel National Center for Personalized Medicine, Weizmann Institute of Science, Rehovot, Israel. [6]Experimental Imaging Center, IRCCS San Raffaele Scientific Institute, Milan, Italy. ✉e-mail: ziv.shulman@weizmann.ac.il

(LZ)[19,20], followed by a transition into the GC dark zone (DZ), where rapid cell proliferation occurs[9,21,22].

DYRK family members are master regulators of proliferation in many cell types[23]. These enzymes are dual-specificity protein kinases that can autophosphorylate their own tyrosines, thereby activating their serine and threonine phosphorylation activity on target proteins[24,25]. Since the DYRK1A locus is located on a region of chromosome 21 that is duplicated in Down syndrome, it is the most extensively studied family member[26,27]. Phosphorylation of c-Myc, c-Jun, Cyclin D1, and Cyclin D3 by DYRK1A labels these proteins for proteasomal degradation, thereby attenuating the rate and magnitude of cell division[23,28–33]. c-Myc and many other DYRK1A targets are expressed in pre-GC B cells and in GC B cells that are selected for enhanced proliferation by their cognate T cells[21,22]. Yet, although DYRK1A is a master regulator of cell-cycle progression, its function in rapidly proliferating GC B cells during an immune response was not examined, and it is not known whether other DYRK1A-mediated mechanisms contribute to the generation of protective antibodies.

In this work, we examine the function of DYRK1A during B-cell immune response to viral infection and to vaccine-derived antigens. We find that DYRK1A is essential for protection from viral infection through CSR, mediated by phosphorylation of MSH6. Furthermore, DYRK1A regulates GC seeding and subsequent effective clonal expansion through the attenuation of cell-cycle progression.

## Results

### Class-switch recombination requires DYRK1A independently of B-cell proliferation

To study the function of DYRK1A in B cells, we crossed mice carrying a conditional inactivation of *Dyrk1a* gene with mice expressing B-cell-specific Cre, under the CD23 promoter (*CD23.Cre.Dyrk1a^{fl/fl}*)[28,34]. B cells derived from littermate control mice (*CD23.Cre.Dyrk1a^{+/+}*) expressed DYRK1A, and its levels were increased in response to LPS stimulation[35], whereas B cells derived from *CD23.Cre.Dyrk1a^{fl/fl}* mice lacked DYRK1A before and after stimulation with LPS (Fig. 1a). To examine whether DYRK1A is involved in B-cell activation and proliferation, splenic B cells were stimulated with LPS for 16 h, and CD86 upregulation, a hallmark of B-cell activation, was examined. Flow cytometry analysis showed that *Dyrk1a*-deficient and control B cells expressed similar levels of CD86 (Fig. 1b). Furthermore, B-cell proliferation was examined through stimulation of cultured CellTrace Violet (CTV)-labeled cells with LPS + IL-4, LPS, or αIgM for 3 days. Stimulation with LPS induced a significant increase in cell division and proliferation of *Dyrk1a*-deficient B cells relative to WT, whereas no effect or decreased proliferation was observed after LPS + IL-4 or αIgM stimulation, respectively (Fig. 1c). Furthermore, we directly investigated CSR in proliferating B cells by stimulation of splenic B cells with LPS in the presence of IL-4, followed by flow cytometry analysis after 3 days. Whereas a population of B cells that expressed IgG1 in response to the stimulation was detected in the control cells, *Dyrk1a*-deficient B cells showed a significantly smaller frequency of IgG1-positive cells (Fig. 1d). Similarly, class-switching to IgG3 or IgG2a/b, which is mediated by LPS stimulation without additional cytokines, was also significantly reduced in *Dyrk1a*-deficient B cells (Fig. 1d). It was previously shown that CSR is linked to cell division[36,37]; however, since *Dyrk1a*-deficient B cells stimulated with LPS + IL-4 showed comparable rather than diminished proliferation in vitro, we conclude that reduced cell division does not explain their CSR defect.

### MSH6 is a direct target of DYRK1A phosphorylation

It was previously shown that DYRK1A phosphorylates c-Myc in acute myeloid leukemia (AML), and Cyclin D3 in pre-B cells to enhance their degradation[28,33]. Using flow cytometry, we did not detect measurable changes in the expression of these cell-cycle regulators in *Dyrk1a-deficient* naïve, GC B cells, or plasma cells compared to controls

(Supplementary Fig. 1). To further explore the mechanism of DYRK1A-mediated CSR and to gain further insights into its possible substrates, stimulated B cells were subjected to global and phosphoproteomic analyses. We identified 103 proteins that were upregulated and 20 proteins that were downregulated in LPS-stimulated B cells, while after αIgM treatment, the expression of 314 and 55 proteins was upregulated or downregulated, respectively (FC > 1.5 or < −1.5, P < 0.05) (Fig. 2a). Using the Metascape tool for the analysis of datasets at the systems level[38], we found that DYRK1A is involved in DNA-damage and repair pathways and in cell-cycle progression. However, the proliferation-related proteins c-Myc and Cyclin D3 were not detected in our proteomic analysis (Fig. 2a, b). Since DYRK1A is a kinase[28,39], we next used phosphoproteomics analysis to screen for specific targets that could potentially explain the defect in CSR. In this analysis, a potential target site was considered as differentially phosphorylated if its phosphorylation level significantly changed in *Dyrk1a*-deficient B cells compared to WT (FC > 1.5 or < −1.5, P < 0.05). Protein sites were excluded if their protein level was significantly changed in the same direction as the phosphorylation level (FC > 1.25 or < −1.25, P < 0.1). This analysis implicated 181 hypophosphorylated sites in LPS-stimulated B cells, and 454 hypophosphorylated sites in αIgM stimulated cells (Fig. 2c). Examination of the biological pathways affected by the loss of protein phosphorylation showed a robust change in DNA recombination, mismatch repair, and cell-cycle-related genes (Fig. 2d). Specifically, the mismatch repair protein MSH6, whose total protein expression was unchanged, showed reduced phosphorylation at five different sites (Fig. 2c, e). Similar findings were detected in LPS + IL-4 stimulated B cells (Supplementary Fig. 2). These observations indicate that MSH6 may be a direct target of DYRK1A.

MSH6 was previously reported to have an important role in B-cell CSR[40–43]. Since *Dyrk1a*-deficient B cells showed a severe defect in this process as well, we examined whether MSH6 is a direct substrate of DYRK1A. Using recombinant DYRK1A and MSH6 proteins, the phosphotransferase activity of DYRK1A was examined in vitro. MSH6 was not phosphorylated in the absence of DYRK1A or ATP, but when both recombinant proteins and ATP were incubated together, MSH6 acquired a phosphorylation signal. Furthermore, DYRK1A was autophosphorylated, as expected[44] (Fig. 2f). To examine if MSH6 phosphorylation promotes CSR, we transduced WT splenic B cells with retroviral constructs encoding WT *MSH6* or a non-phosphorylatable *MSH6* mutant (T326A) (Fig. 2c, e). Transduced B cells were stimulated with LPS and IL-4 for 3 days and examined by flow cytometry for IgG1 class-switching. Although the results of this assay were quite variable, the frequency of IgG1 class-switched *Msh6T326A* transduced B cells was significantly reduced compared to the control (Fig. 2g). It is important to note that endogenous WT MSH6 was also expressed in the B cells that were transduced with *MSH6 T326A*, and additional DYRK1A phosphorylation sites on MSH6 might also have a function in CSR. Nonetheless, the reduction in class-switching was significant, demonstrating that phosphorylation of MSH6 at the DYRK1A-targeted site is required for intact CSR.

### B-cell class switch recombination in vivo requires DYRK1A

To examine whether DYRK1A regulates CSR in vivo, the presence of class-switched antibodies in the serum of *CD23.Cre.Dyrk1a^{fl/fl}* mice was examined by ELISA. Serum immunoglobulins derived from unmanipulated mice showed reduced titers of class-switched antibodies in *CD23.Cre.Dyrk1a^{fl/fl}* mice, including IgA, IgG1, and IgG2b, whereas IgM titers were unchanged compared with control mice (Fig. 3a). Nonetheless, the total number of bone marrow (BM) PCs in *Dyrk1a*-deficient mice was similar to the controls, suggesting that a defect in the generation of these cells cannot explain the lack of class-switched antibodies in the mouse sera (Fig. 3b). Intracellular staining for Ig isotypes indicated a significant reduction in IgG1[+] but not IgA[+] PCs in the BM of *CD23.Cre.Dyrk1a^{fl/fl}* mice (Fig. 3b and

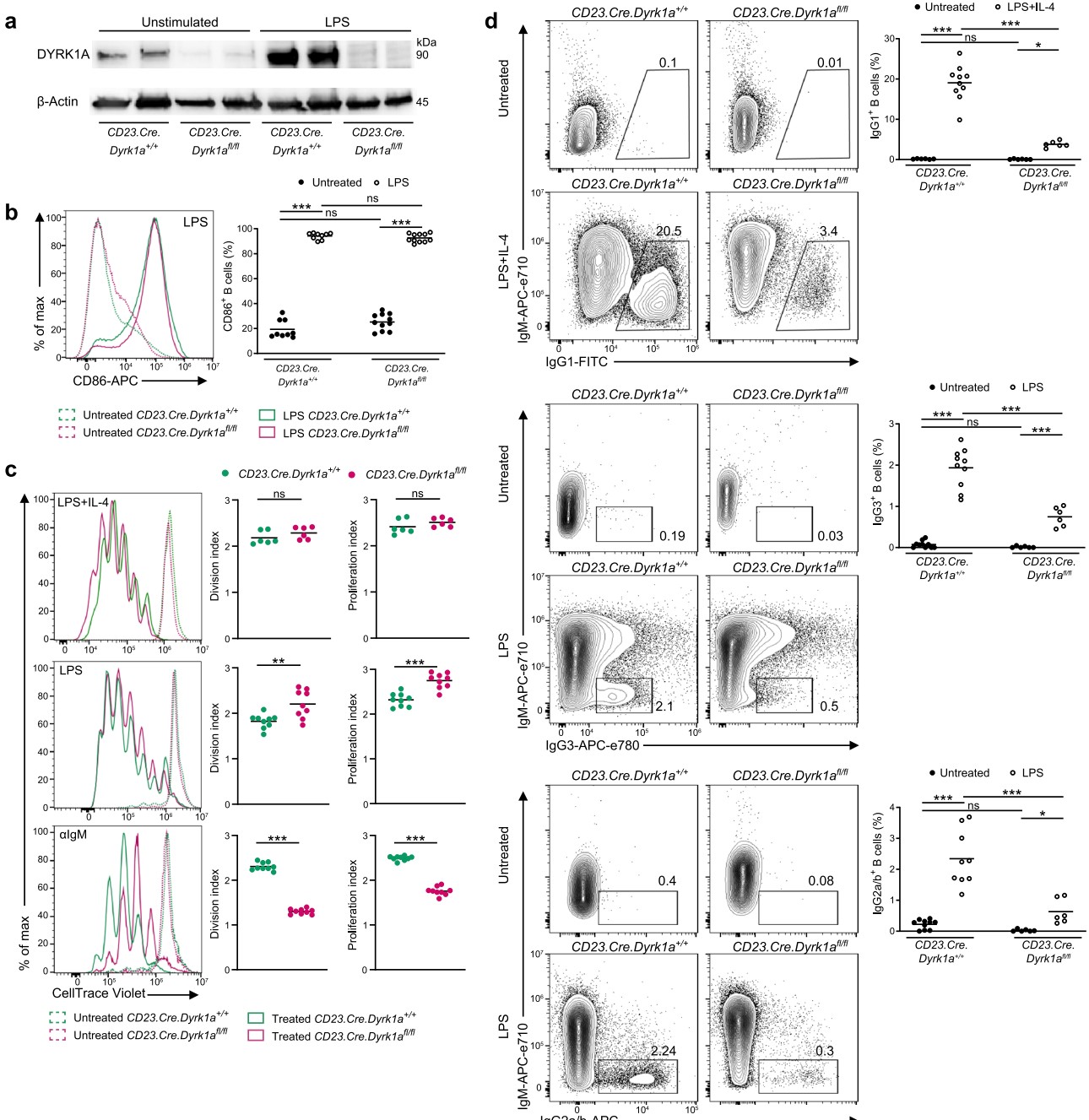

**Fig. 1 | Class-switch recombination requires DYRK1A independently of B-cell proliferation. a** DYRK1A protein expression was determined by western blot analysis of B cells that were either left unmanipulated or stimulated with LPS for 3 days. Blots show two independent biological repeats. **b** Representative flow cytometry histograms and frequencies of activated B cells 16 h after LPS stimulation in vitro ($n = 9–11$; three independent experiments, one-way ANOVA). **c** Representative flow cytometry histograms and quantification of CellTrace Violet dilutions representing the proliferation of splenic LPS + IL-4, LPS or αIgM treated B cells for 3 days ($n = 6–9$; three independent experiments, two-tailed Student's $t$ test). **d** Representative flow cytometry plots and frequencies of class-switched B cells derived from naive spleens that were either left unmanipulated or stimulated in vitro with LPS + IL-4 or LPS for 3 days ($n = 9–11$; three independent experiments, one-way ANOVA). Each dot in the graphs represents a single mouse; *$P = 0.05$, **$P \leq 0.01$, ***$P \leq 0.001$; ns not significant.

Supplementary Fig. 3). These results indicate that DYRK1A has a function specifically in the generation of IgG class-switched antibodies, rather than in PC formation.

The GC reaction is the major source of class-switched PCs. To examine whether the reduced frequency of class-switched PCs in *CD23.Cre.Dyrk1a*$^{fl/fl}$ mice is a result of an impaired GC response, *CD23.Cre.Dyrk1a*$^{fl/fl}$ and littermate mice were immunized subcutaneously in the hind footpads with hapten (4-hydroxy-3-nitrophenyl [NP]) coupled to keyhole limpet hemocyanin (KLH) in alum.

Flow cytometric analysis 7 days later, showed that the frequency of GC B cells was not altered in the absence of DYRK1A (Fig. 3c). Nonetheless, and consistent with the findings we described under homeostatic conditions, the frequency of IgG1⁺ class-switched B cells in *Dyrk1a*-deficient GCs was significantly lower compared to the controls (Fig. 3c). Since CSR primarily occurs prior to the establishment of mature GCs[10], we conclude that DYRK1A functions in the generation of class-switched B cells at early stages of the immune response.

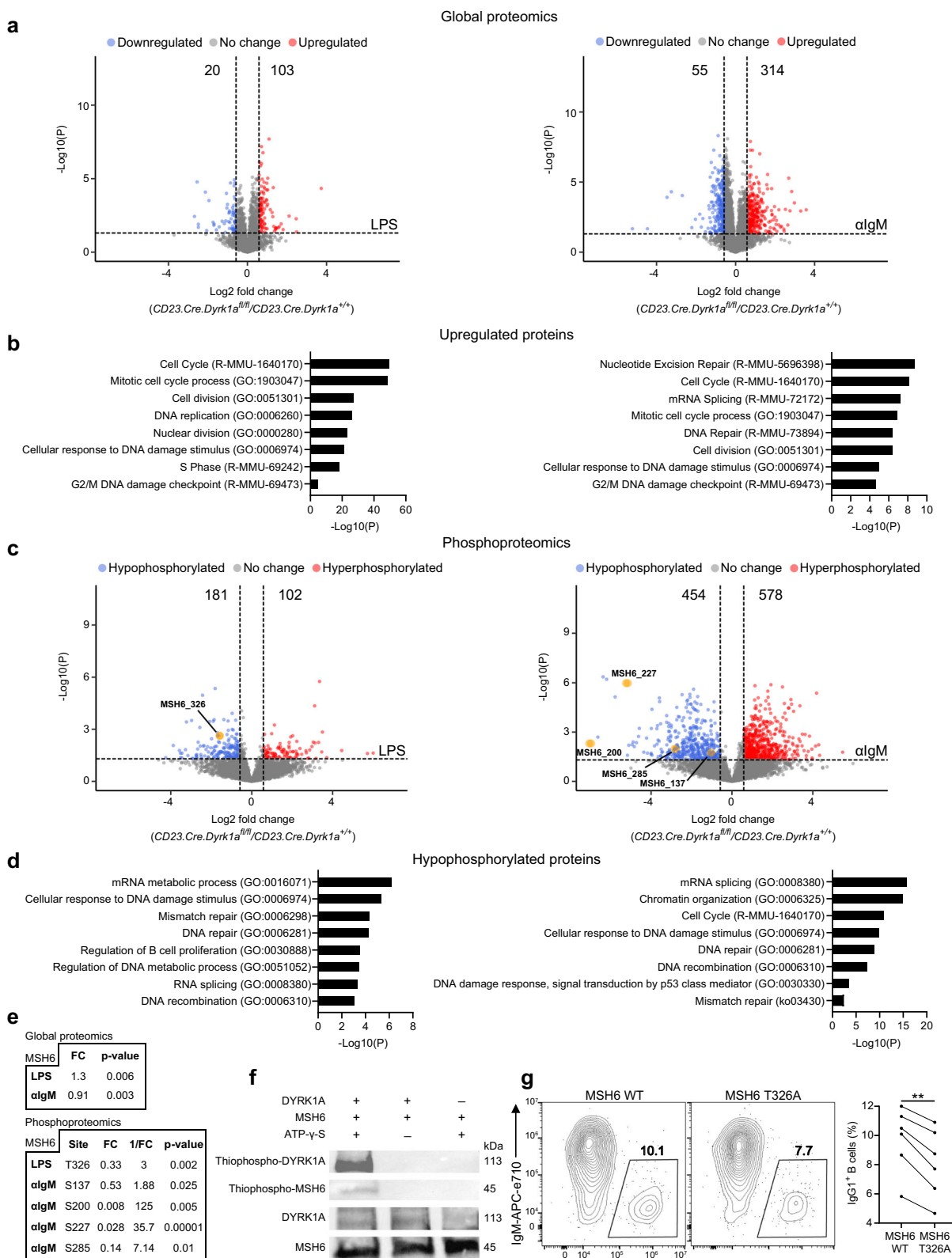

**Fig. 2 | MSH6 is a direct target of DYRK1A phosphorylation. a, b** Volcano plots depicting changes in protein expression in *Dyrk1a*-deficient B-cell mice compared to WT littermates, which were both stimulated with LPS or αIgM for 3 days (**a**); biological pathway analysis was performed on upregulated proteins (**b**); (hypergeometric test and Benjamini–Hochberg P value correction by Metascape). **c, d** Volcano plots showing changes in specific phosphorylation sites (**c**); biological pathway analysis was performed on hypophosphorylated sites in B cells stimulated with LPS or αIgM for 3 days (**d**); (*n* = 5; three independent experiments, hypergeometric test and Benjamini–Hochberg P value correction by Metascape).

Colored points correspond to P value <0.05 and log2 FC > 0.58 (red) or < −0.58 (blue). **e** Table listing detected MSH6 global (top) and phosphoproteomic (bottom) changes (two-tailed Student's *t* test). **f** Representative western blot showing an in vitro direct kinase assay (two independent experiments). **g** Representative flow cytometry plots and frequencies of IgG1+ WT B cells transduced with either MSH6 WT or *MSH6 T326A* retroviral constructs, and stimulated in vitro with LPS + IL-4 for 3 days (*n* = 6; two independent experiments, two-tailed paired Student's *t* test). The connected dots represent data from the same mouse; **P ≤ 0.01.

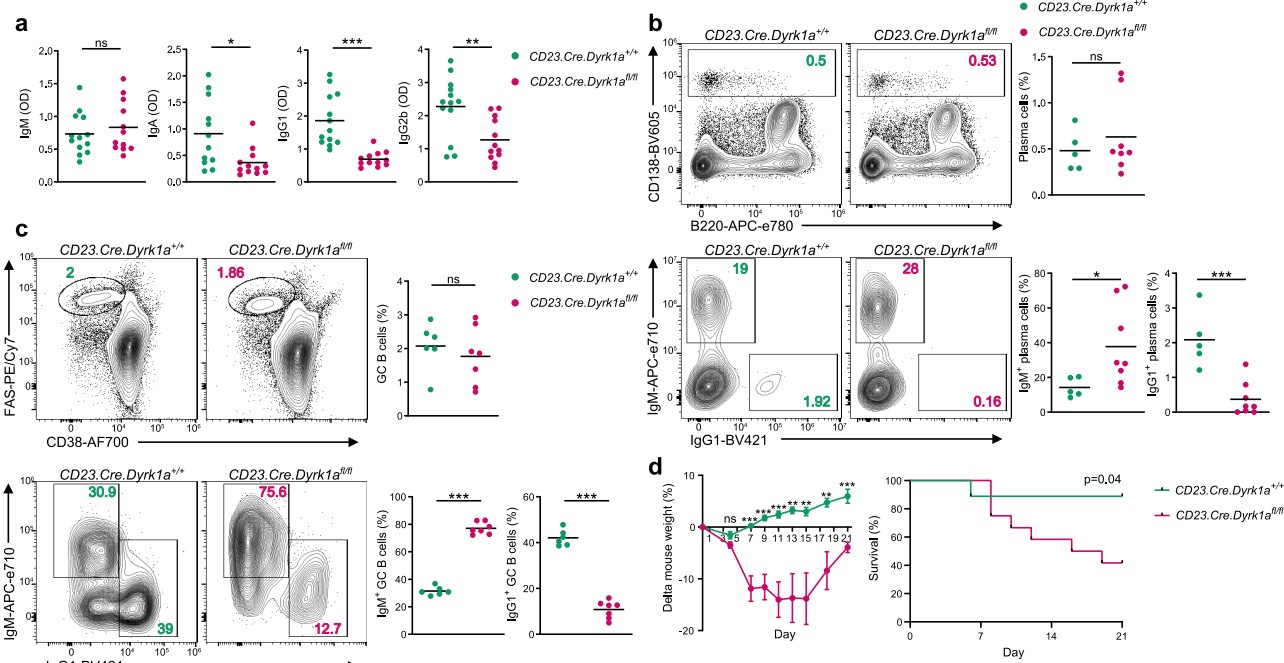

**Fig. 3 | B-cell class switch recombination in vivo requires DYRK1A. a** Serum IgM, IgA, IgG1, and IgG2b titers as determined by ELISA ($n = 12$–13 unmanipulated mice; two independent experiments, two-tailed Student's $t$ test). **b** Representative flow cytometry plots and frequencies of total and IgG1+ BM PCs in unmanipulated mice ($n = 5$–8; three independent experiments, two-tailed Student's $t$ test). **c** Representative flow cytometry plots and frequencies of total GC and isotype-specific GC B cells in popliteal LNs 7 days after NP-KLH immunization ($n = 6$–7; two independent experiments, two-tailed Student's $t$ test). **d** Time course of changes in mouse weight and survival following i.v. VSV-Ind infection ($n = 9$–12 mice; two independent experiments, multiple two-tailed Student's $t$ tests, and the log-rank Mantel−Cox test). Each dot in the graphs represents a single mouse; *$P = 0.05$, **$P \leq 0.01$, ***$P \leq 0.001$, ns not significant.

To understand whether DYRK1A in B cells has a protective function during pathogen invasion, *CD23.Cre.Dyrk1a*<sup>fl/fl</sup> and control mice were infected with Vesicular Stomatitis Indiana Virus (VSV-Ind), which is cleared from the host through an antibody-mediated immune response[3]. After 1 week following systemic infection, significant weight loss was observed among the *CD23.Cre.Dyrk1a*<sup>fl/fl</sup> mice, whereas the control mice recovered by day 7 post-infection, and their body weight started to increase. By day 21 post-infection, 7 out of 12 (58%) *CD23.Cre.Dyrk1a*<sup>fl/fl</sup> mice died, while only 1 out of 9 (11%) control mice died (Fig. 3d). The mice that lost weight but eventually recovered and survived the viral infection had detectable VSV-specific IgG antibodies in their sera, suggesting that they overcame the infection through a compensatory mechanism that supports CSR (Supplementary Fig. 4). We conclude that DYRK1A in B cells has an important function in mounting a protective immune response against viral infection.

### DYRK1A restricts germinal center seeding by antigen-specific B cells

MSH6 is important for both the proper acquisition of CSR, which primarily takes place after the initial B-cell activation, and partially for somatic hypermutation (SHM), occurring during the GC reaction. To examine the function of DYRK1A in SHM, we first examined the GC response in immunized mice. To overcome the CSR defect, we crossed *Dyrk1a*<sup>fl/fl</sup> mice with *Aicda*<sup>Cre/+</sup> and *Rosa26*<sup>flox-stop-flox-tdTomato</sup> mice. AID (encoded by *Aicda*) is upregulated during initial B-cell activation, after T cell-dependent antigen encounter, and prior to GC seeding[10,45–47]. Since CSR occurs after AID expression but before GC formation, this model allowed us to bypass the early defects and examine class-switched B cells in the GC response. In contrast to the results obtained using the *CD23.Cre* model, under homeostasis, increased IgG1 titers were detected in *AID.Cre.Dyrk1a*<sup>fl/fl</sup> mice, while IgA and IgG2b titers did not change significantly (Fig. 4a). While the overall frequency of PCs in the BM of *AID.Cre.Dyrk1a*<sup>fl/fl</sup> mice was not altered, a twofold increase in

IgG1+ PCs was detected in these mice compared to control animals (Fig. 4b). Thus, we conclude that DYRK1A has a critical function in CSR in the early response, while it does not play a role in later events, such as in the maintenance of class-switched PCs.

Since most of the class-switched PCs originate from the GC, we examined whether changes occur in this compartment in immunized *AID.Cre.Dyrk1a*<sup>fl/fl</sup> mice. Flow cytometric analysis 7 days after immunization with NP-KLH, demonstrated that the frequency of GC B cells and PCs in the draining LNs of *AID.Cre.Dyrk1a*<sup>fl/fl</sup> mice was significantly increased compared to littermate controls (Fig. 4c). Accordingly, more IgG1+ NP-specific B cells were detected on day 7 of the response, demonstrating that an increased antigen-specific immune response occurs at this time point (Supplementary Fig. 5). Furthermore, since CSR occurs prior to GC formation but after AID expression, no defect in the frequency of IgG1+ GC B cells was observed (Fig. 4c). The increased frequency of GC B cells detected by flow cytometry could indicate either larger GC compartments or an increased number of individual GCs. To resolve this issue, we used intravital two-photon laser scanning microscopy (TPLSM) to image the popliteal LNs of immunized control and *AID.Cre.Dyrk1a*<sup>fl/fl</sup>.*Rosa26*<sup>flox-stop-flox-tdTomato</sup> mice. This approach implied that the size of each individual GC was significantly larger in the *AID.Cre.Dyrk1a*<sup>fl/fl</sup> compared to control mice (Fig. 4d). Nonetheless, the GC size in *AID.Cre.Dyrk1a*<sup>fl/fl</sup> mice was reduced to normal levels on day 14 of the response and was slightly smaller compared to the control mice 21 days after the immunization (Supplementary Fig. 6). These findings suggest that DYRK1A restricts GC seeding during early phases of the response, but not at later time points.

To investigate whether the elevated size of GCs that we observed after immunization with a model antigen is reproducible in an infection model, we infected mice with VSV-Ind. Similar to the vaccination model, 7 days after VSV-Ind injection to the footpad of *AID.Cre.Dyrk1a*<sup>fl/fl</sup>, the frequency of GC B cells in the draining popliteal LNs was higher

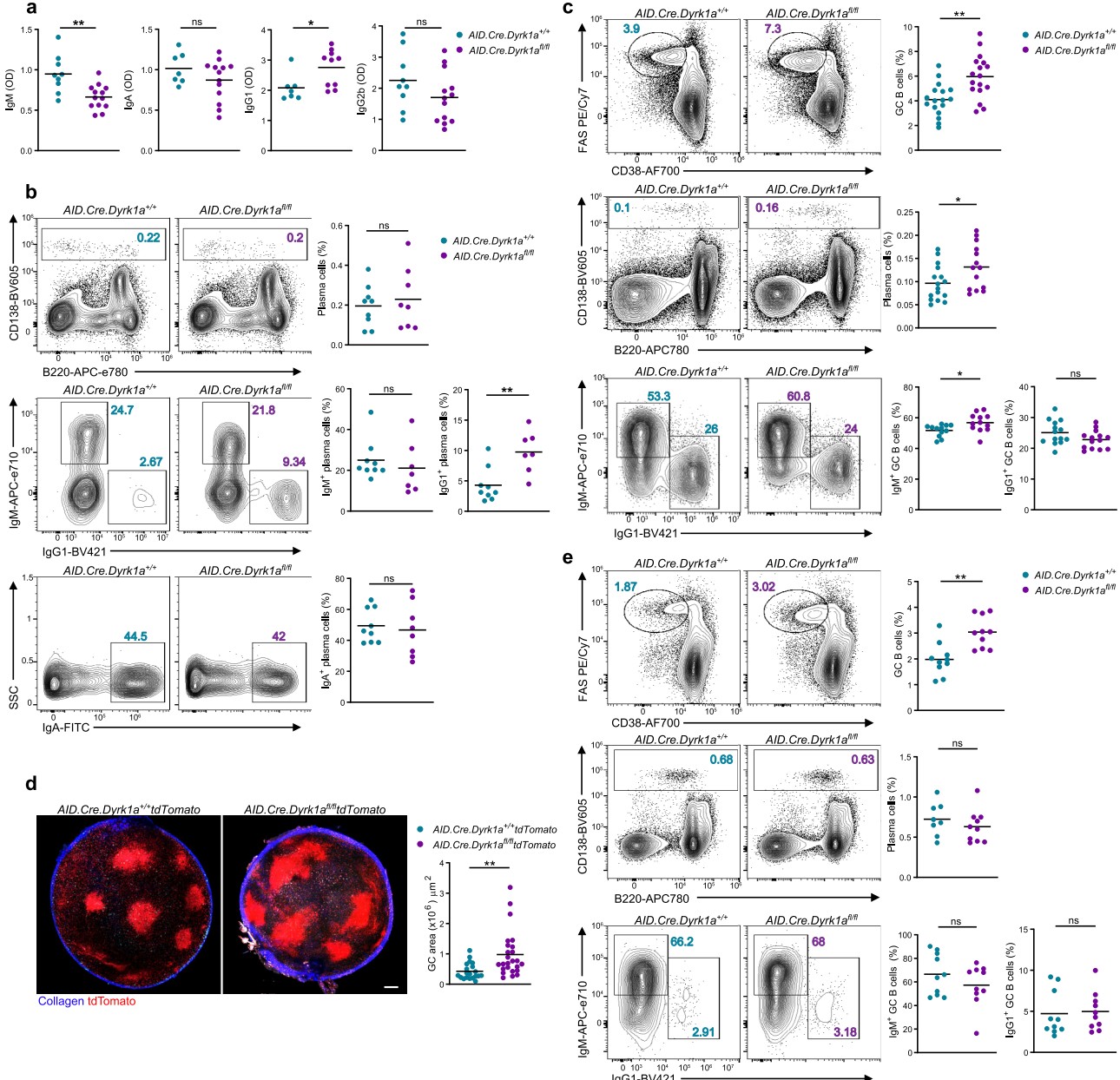

**Fig. 4 | DYRK1A restricts the magnitude of the B-cell immune response.**
**a** Serum IgM, IgA, IgG1, and IgG2b titers as measured ELISA (*n* = 7–13 unmanipulated mice; two independent experiments, two-tailed Student's *t* test).
**b** Representative flow cytometry plots and frequencies of total and class-switched PCs in the BM of unmanipulated mice (*n* = 7–9; two independent experiments, two-tailed Student's *t* test). **c** Representative flow cytometry plots and frequencies of total GC, PC, and isotype-specific GC B cells in popliteal LNs 7 days after NP-KLH immunization (*n* = 12–17; three independent experiments, two-tailed Student's *t* test). **d** Representative TPLSM images of popliteal LN-

derived from *AID.Cre.Dyrk1a*$^{fl/fl}$*.Rosa26*$^{flox-stop-flox-tdTomato}$ mice and quantification of GC area 7 days after NP-KLH immunization. Each dot in the graph represents a single GC (*n* = 3–5; three independent experiments, two-tailed Student's *t* test); scale bar 200 μm. **e** Representative flow cytometry plots and frequency quantification of total GC, PC, and isotype-specific GC B cells in popliteal LNs, 7 days after VSV-Ind infection (*n* = 8–10; two independent experiments, two-tailed Student's *t* test). Each dot in the graphs represents a single mouse; *\*P* = 0.05, *\*\*P* ≤ 0.01, *\*\*\*P* ≤ 0.001, ns not significant.

compared to control mice, though the frequency of PCs and class-switched B cells did not change in this setting (Fig. 4e). Based on these results, we conclude that after initial B-cell activation and CSR, DYRK1A restrains GC seeding during early stages of the response to vaccination or virus infection.

## DYRK1A is required for proper B-cell clonal expansion in germinal centers

*Msh6*-deficient mice have altered SHM patterns in their B-cell immunoglobulin genes[41]. To examine if DYRK1A has a function in clonal

expansion and insertion of SHM in the GC reaction, we sorted single IgG1[+] GC B cells from the LNs of NP-KLH immunized mice, followed by *Ighg1* amplification and sequencing[48]. This analysis showed that the magnitude of clonal expansion and diversity in control versus *Dyrk1a*-deficient mice was similar to wild type both 7 and 21 days post-immunization (Fig. 5a), as was the specific V-region usage (Fig. 5b). *Dyrk1a*-deficient B cells acquired SHM in their variable region to a lesser extent compared to the control mice, but although this difference was statistically significant, it was relatively small (Fig. 5c). Furthermore, lineage tree reconstruction of representative expanded

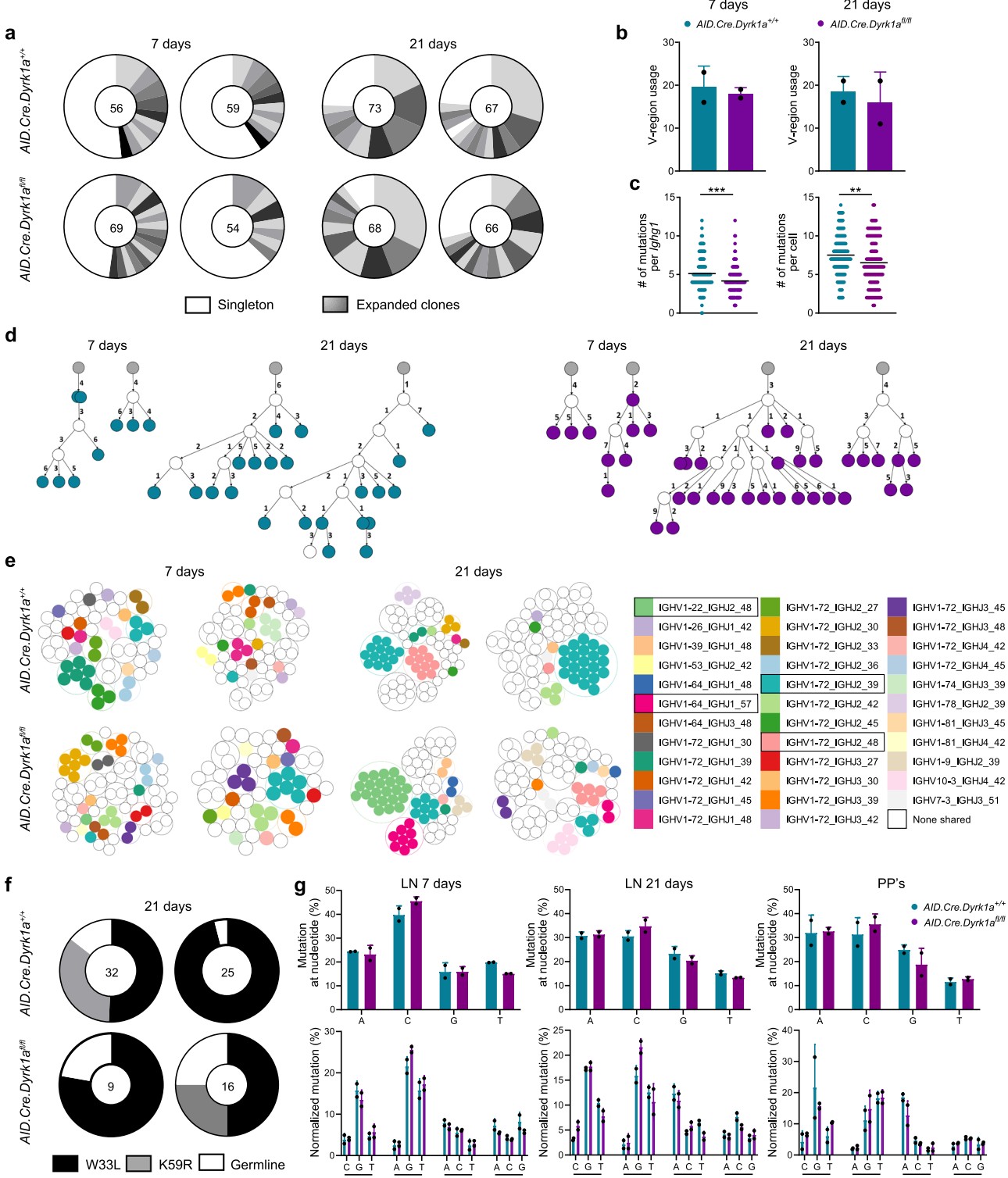

**Fig. 5 | Antigen-specific clonal expansion but not SHM depends on DYRK1A functions. a** Pie charts showing the clonal distribution of *Ighg1* sequences in GC B cells derived from one LN of a single mouse, 7 or 21 days after NP-KLH immunization. Each segment represents a unique clone. The total number of analyzed sequences is indicated in the center of each chart (*n* = 2; two independent experiments). **b** The number of different V-regions detected in *Ighg1* sequences as in (**a**). Bars and error bars represent the mean with SD (*n* = 2; two independent experiments). **c** The number of SHMs per analyzed sequence as in (**a**). (*n* = 113–135; two independent experiments, two-tailed Student's *t* test); \*\**P* ≤ 0.01, \*\*\**P* ≤ 0.001. **d** Phylogenetic trees of representative individual clones. Gray circles represent the hypothetical germline configuration. White circles represent hypothetical ancestors. Numbers appearing next to the arrows represent the number of distinct

mutations accumulated between clonal members. **e** Shared clone analysis of the sequences as in (**a**). Each color represents an individual shared clone. White represents clones detected in one LN and not in the others (non-shared). **f** The frequency of single cells bearing the NP-specific high-affinity mutations in the *Ighv1-72* heavy chain gene, 21 days after immunization. The total numbers of analyzed *Ighv1-72* sequences are indicated in the center of each chart. **g** Analysis of the mutational landscape showing total mutations at each base position and individual nucleotide substitutions in *Ighg1* sequences from GC B cells derived from one LN of a single mouse 7 or 21 days after NP-KLH immunization, or a single PP. Bars and error bars represent the mean with SD (*n* = 2; two independent experiments); \*\**P* ≤ 0.01, \*\*\**P* ≤ 0.001.

clones showed that *Dyrk1a*-deficient B cells diversify and accumulate SHMs in the GC, suggesting that the GC reaction supports antibody-affinity maturation (Fig. 5d). To further understand whether DYRK1A has a function in affinity maturation and selective antigen-specific clonal expansion, we specifically inspected *IGHV1-72*, which is the typical clone that responds to NP in C57BL/6 mice[49]. For this purpose, we used shared clone analysis which demonstrates clonal expansion of specific V-J *Ighg1* sequences. Both control and *AID.Cre.Dyrk1a^{fl/fl}* hosted individual GC B cells bearing *IGHV1-72* 7 days after the immunization. In control mice, the presence of these clones increased after an additional 14 days, demonstrating clone-specific expansion in these GCs. In contrast, in *AID.Cre.Dyrk1a^{fl/fl}* mice, *IGHV1-72* B-cell clones were not clonally expanded, and a clone that carries a different BCR, *IGHV1-22*, dominated the GC response in one mouse, whereas in the second mouse, no clonal expansion of *IGHV1-72* or of other shared clones was observed (Fig. 5e). To examine if the defect in clonal expansion of antigen-specific B cells that carry *IGHV1-72* is a result of failure in the selection of B cells that acquired affinity-enhancing SHMs, we quantified the frequency of two mutations that are associated with increased immunoglobulin affinity, W33L and K59R. The presence of these mutations was comparable in littermates and *AID.Cre.Dyrk1a^{fl/fl}* mice, suggesting that B-cell selection is not impaired in *Dyrk1a*-deficient GC B cells (Fig. 5f). We conclude that DYRK1A is required for proper clonal expansion of antigen-specific B cells that carry high-affinity BCRs during the GC reaction.

## DYRK1A is not required for nucleotide base substitutions during somatic hypermutation

We demonstrated that DYRK1A regulates CSR through phosphorylation of MSH6 early in the response, and that this kinase also controls the magnitude of the GC reaction at later time points. In addition to its role in CSR, MSH6 was suggested to modulate the pattern of specific SHM in GCs[41,50]. To examine whether DYRK1A controls B-cell functions in the GC through MSH6 activity, we further investigated the mutational landscape of *Ighg1* in *AID.Cre.Dyrk1a^{fl/fl}* mice. For this purpose, we re-examined the *Ighg1* sequences to identify specific nucleotide substitutions that might be affected by the lack of MSH6 functions. Furthermore, since this analysis was previously done using immunoglobulin sequences derived from PPs, we also sequenced *Ighg1* from PP GCs of 1-year-old control and *AID.Cre.Dyrk1a^{fl/fl}* mice. Analysis of specific changes in the mutational patterns showed no differences in nucleotide substitutions in *Dyrk1a*-deficient B cells compared to the control mice in either draining LNs or PPs (Fig. 5g). A similar analysis of SHM upstream of the core Sμ-region done using LPS + IL-4 stimulated B cells for 3 days, showed an increase in G/C nucleotide substitutions, as was previously shown in *Msh6*-deficient mice[41,51]. However, since A/T substitutions were observed in *Msh6*-deficient mice and since in this assay, the mutation frequency is very low, the number of mutated sequences was not sufficient for statistical analysis (Supplementary Fig. 7). These results suggest that SHMs in the VDJ seqeunces are not regulated through DYRK1A and MSH6 phosphorylation in the GC.

## DYRK1A is a negative regulator of cell-cycle progression in GC B cells

The function of DYRK1A in the regulation of cell-cycle events was shown previously in tumor, neuronal, fibroblastic, and pre-B cells[23,28,30,31,52]. Thus, we examined the possibility that DYRK1A controls GC size by modulating the cell-cycle progression of B cells during the early stages of the immune response. The GC is composed of two zones, the DZ, where B cells undergo clonal expansion and diversify their immunoglobulins by SHM, and the LZ, where B cells are subjected to selection based on their BCR affinity[9,53]. We found that GCs in the draining popliteal LNs of immunized *AID.Cre.Dyrk1a^{fl/fl}* mice exhibit slightly smaller DZ compartments compared to their control

counterparts (Fig. 6a). Nonetheless, the total number of antigen-specific B cells a week after immunization was fourfold higher in *AID.Cre.Dyrk1a^{fl/fl}* mice compared to the controls, demonstrating that more cells take part in the GC reaction regardless of zonation (Supplementary Fig. 5). To examine whether DYRK1A controls gene expression profiles and biological pathways that may explain the observed changes in GC size, we sorted LZ and DZ B cells and subjected them to RNA-seq analysis. We found in the DZ 747 genes that were upregulated, and 516 genes that were downregulated when compared to control cells (log2 FC ≥ ± 0.58, adjusted *P* < 0.05). A change in expression of only 28 genes was detected in the LZ, suggesting that DYRK1A plays a minor role, if any, in this GC compartment. (Fig. 6b). Utilizing the Metascape tool to identify the biological processes associated solely with the upregulated gene profile of DZ B cells, we found that in the absence of DYRK1A, cell cycle, DNA replication, nuclear division and DNA repair events are massively altered (Fig. 6c), providing an explanation for the enlarged GC size we observed (Fig. 4). To identify signaling pathways that were modified in *Dyrk1a*-deficient DZ B cells, we analyzed the gene signatures by gene set enrichment analysis (GSEA). This analysis showed significant changes in genes related to the G2M and E2F pathways, known as entry checkpoints to DNA synthesis and phases of mitosis[54,55] (Fig. 6d). To determine whether the detected cell division transcriptomic changes indeed affect the B-cell cycle, we injected immunized mice with the nucleoside analog EdU, which is incorporated into newly formed DNA in dividing cells. Flow cytometric analysis of the DZ and LZ compartments, indicated that the fraction of both total EdU incorporating cells, and specific cells in the S-phase, was slightly higher in *Dyrk1a*-deficient DZ B cells (Fig. 6e, f). Enhanced proliferation and initiation of cell cycle, were detected in DZ B cells and not in LZ B cells following dual-labeling with EdU and BrdU (Supplementary Fig. 8). Thus, the enhanced B-cell proliferation in the DZ primarily affects the global GC size rather than a specific zone within it. Furthermore, we conclude that DYRK1A limits the number of antigen-specific B cells during the early stages of the GC reaction, and is required for their proper expression of cell-cycle genetic programs.

## Discussion

CSR is critical for the generation of antibodies with the capacity to clear infected or malignant cells, by triggering immune cell effector function[5]. Here, we describe a previously unknown molecular mechanism that controls CSR recombination at the post-translational level through the functions of DYRK1A. Whereas this kinase is a master regulator of cell proliferation through targeting many cell-cycle-related proteins for degradation, we demonstrate an additional function for DYRK1A in CSR and suggest that MSH6 is one of its major targets.

Single base nucleotide mismatches are recognized by the MutSα complex, which is composed of mismatch repair (MMR) proteins including MSH2, and either the MSH3 or MSH6 subunits[56–58]. The recruitment of this complex to the switch region leads to the excision of the mutated sites followed by DNA resynthesis by an error-prone polymerase[59,60]. Generation of mismatched nucleotides through AID enzymatic activity followed by DNA repair mechanisms is essential for CSR[42]. *Msh2*- *Msh6*- but not *Msh3*-deficient B cells are defective in this process, and indeed, MSH3 was not detected as a target for DYRK1A in our phosphoproteomics analysis, indicating that DYRK1A is not involved in regulating its functions[41,51,61–65]. In addition to DYRK1A, regulation of MSH6 functions by phosphorylation through the activity of PKC and casein kinase activity was previously demonstrated in other cell types[58]. However, the cells in that study did not express AID, which is unique to B cells, nor other components that specifically support CSR[11]. DYRK1A was previously linked to DNA-damage responses through phosphorylation of SIRT1 and deacylation of P53, and this mechanism is required for cell survival[66]. In contrast, we show that

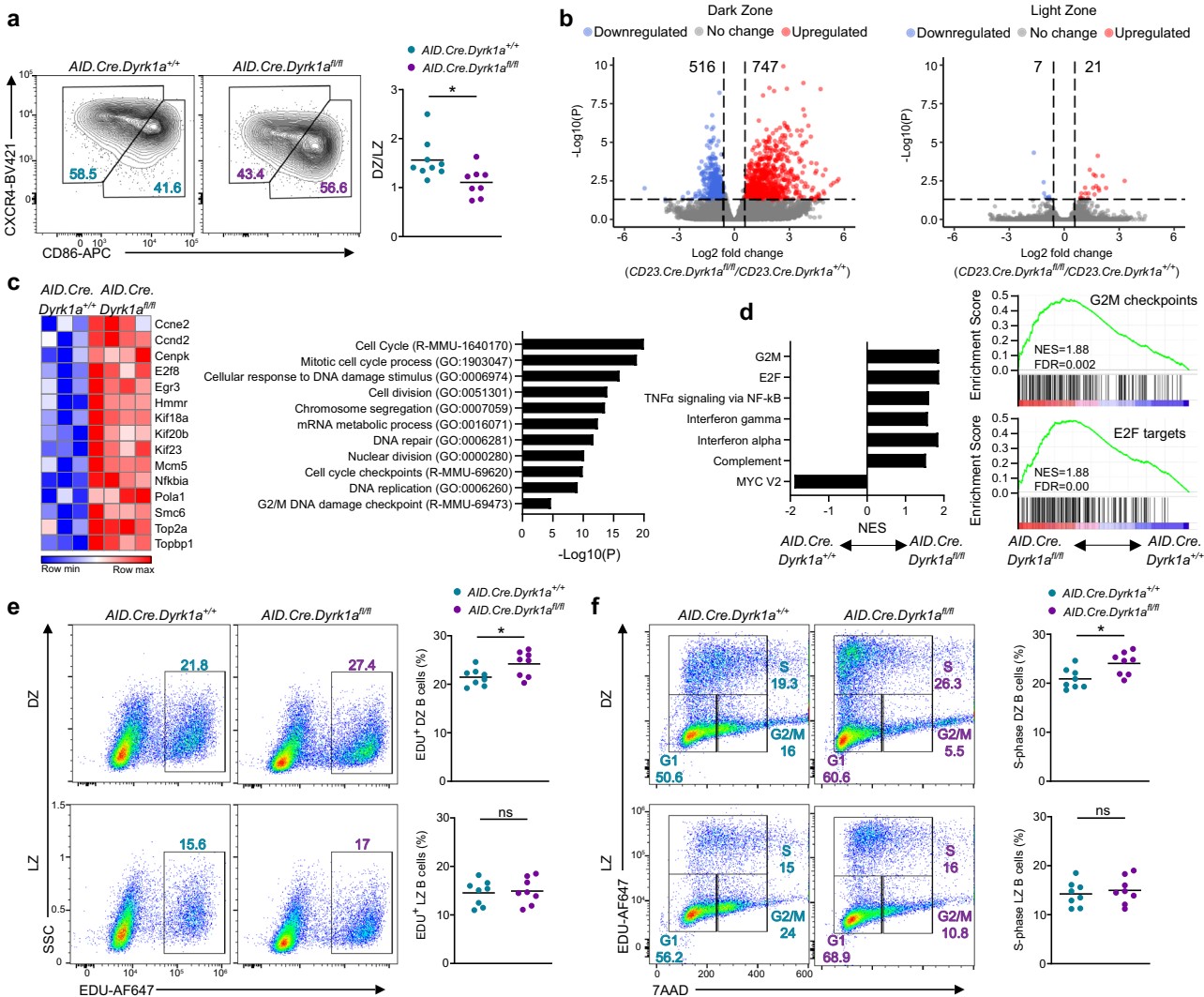

**Fig. 6 | DYRK1A is a negative regulator of cell-cycle progression in GC B cells.**
**a** Representative flow cytometry plots and ratio of DZ to LZ GC B cells in popliteal LNs 7 days after NP-KLH immunization ($n = 8–9$; four independent experiments, two-tailed Student's $t$ test). **b** Volcano plots showing differential gene expression in the LZ and DZ of GC B cells derived from mice, as in (**a**) ($n = 3–4$; two independent experiments). Colored points correspond to adjusted $P$ value <0.05, and log2 FC > 0.58 (red) or < −0.58 (blue). **c** Heatmap of differentially expressed genes and biological pathway analysis of upregulated genes of DZ B cells, as in (**a**)

(hypergeometric test and Benjamini–Hochberg $P$ value correction by Metascape). **d** GSEA analysis of upregulated DZ B-cell gene expression profiles from mice as in (**a**). NES, normalized enrichment score; FDR, false discovery rate. **e**, **f** Analysis of the different cell-cycle stages in DZ and LZ B cells by EdU incorporation and 7AAD DNA staining, 7 days after NP-KLH immunization ($n = 8$; three independent experiments, two-tailed Student's $t$ test). SSC, side scatter. Each dot in the graph represents a single mouse; *$P = 0.05$, **$P ≤ 0.01$, ***$P ≤ 0.001$, ns not significant.

DYRK1A deficiency enhances cell proliferation when cells are subjected to DNA damage by AID, suggesting that P53-mediated cell survival mechanisms do not have a function in GC B cells. The MutSα complex induces the insertion of additional SHMs into the immunoglobulin locus after the generation of nucleotide mismatches by AID activity[41,50]. However, as opposed to CSR, DYRK1A-mediated phosphorylation did not have a major function in this process. This suggests that a different overlapping molecular mechanism compensates for MSH6 functions in GC B cells, or that the chronic nature of the GC response allows accumulation of SHM through a different pathway[50]. Whether MSH6 phosphorylation by DYRK1A is required for specific targeting of the IgH switch regions, and whether it regulates the generation or resolution of double-strand DNA breaks remains to be determined.

In contrast to CSR, the role of DYRK1A in the attenuation of cell-cycle progression has been extensively described[24], but its function in B-cell immune responses was not examined. A previous study demonstrated that DYRK1A is involved in B-cell development through phosphorylation of Cyclin D3[28]. DYRK1A also phosphorylates Myc, a

target that is involved in B-cell selection for clonal expansion by T cells in the GC[32,33]. Although cell proliferation pathways were affected and enhanced in *Dyrk1a*-deficient B cells, no defect in c-Myc and Cyclin D3 protein levels was detected. The rate of B-cell proliferation in the DZ of *AID.Cre.Dyrk1a*$^{fl/fl}$ mice was higher compared to control animals, suggesting that a defect in the pathways restraining the cell cycle in this zone by DYRK1A can explain the increase in global GC size. Lack of proper cell cycle control is expected to lead to an aberrant clonal expansion, and indeed, antigen-specific *Dyrk1a*-deficient B-cell clones that carry an affinity-enhancing mutation in their immunoglobulin genes were unable to expand in the GC over time at the expense of other clones. An additional possibility explaining the increase in GC size is enhanced infiltration of newly activated B cells into the GC[67,68]. However, we did not detect a strong reduction in SHM accumulation in *Dyrk1a*-deficient B cells during the GC response, suggesting that this is not the major explanation for the inability of the antigen-specific clones to dominate the GC reaction over time. More proliferating antigen-specific B cells were detected at the early GC reaction,

indicating that this is the reason for the increase in GC size at this time point. BCL-2 family proteins BCL-XL and MCL-1 reduce B-cell apoptosis in the GC, and its enhanced expression is associated with increased GC size[69,70]. Although we did not detect changes in the expression of BCL-2 family members by RNA-seq, a role for DYRK1A in regulating apoptosis in the GC cannot be excluded. Unexpectedly, the GC size in the AID.Cre model was increased, whereas this effect was not observed in the CD23.Cre mouse model. GC formation and maintenance depend on antigen retention on follicular dendritic cells as immune complexes for long periods[71]. The reduction in IgG antibodies in the CD23.Cre mouse model may affect the ability of *Dyrk1a*-deficient B cells to seed large GCs, a defect that does not occur in the AID.Cre mouse model. Thus, we conclude that in the GC, DYRK1A is a master regulator of cell-cycle progression in the DZ, which maintains proper GC size.

Our results show that MSH6 is a target for DYRK1A, yet a small *Dyrk1a*-deficient, IgG1⁺-expressing B-cell population was detected in our experiments. Furthermore, a significant but relatively modest reduction in CSR was observed upon the removal of a single phosphorylation site from MSH6. These results suggest that other DYRK1A-regulated pathways and other target sites must play a role as well.

Collectively, this study demonstrates two major functions for DYRK1A in B-cell immune responses, which include control of CSR through MSH6 phosphorylation and regulation of cell cycle in GC B cells through multiple cell-cycle factors. Several inhibitors for DYRK1A were developed for the treatment of various types of cancer[24,72]; however, our results suggest that these could enhance the proliferation of malignancies originating in the GC. Nonetheless, suppression of DYRK1A by these inhibitors might be useful for the attenuation of CSR in autoimmune diseases and allergies.

## Methods

### Mice
The research complies with all relevant ethical regulations and all experiments on mice were approved by the Weizmann Institute Animal Care and Use Committee (IACUC number 04240521-2). Dyrk1a^flox/flox, Aicda^Cre/+ and Rosa26^flox-stop-flox-tdTomato mice were purchased from the Jackson Laboratory. CD23^cre mice were generated and provided by M. Busslinger (Research Institute of Molecular Pathology, Vienna, Austria). In all of the experiments, either male or female C57BL/6 mice were used at the age of 6–12 weeks. Mice housing conditions were 12/12 h light/dark cycles at 22 °C ambient temperature with 50% humidity.

### Immunizations and infections
Mice were injected with 25 μl PBS containing 10 μg NP-KLH (BioSearch Technologies) in alum into each hind footpad. For Vesicular Stomatitis Indiana Virus (VSV-Ind) infections, mice were either injected with 10⁶ PFU i.v. or with 10⁵ PFU into each hind footpad.

### Flow cytometry
Popliteal LNs were removed, washed in cold PBS, and forced through a 70-μm mesh into PBS containing 2% FCS and 1 mM EDTA to create single-cell suspensions. Cells were subsequently incubated with fluorescently labeled antibodies (Table 1) for 30 min on ice. Intracellular antibody staining was performed after fixation and permeabilization with Fixation/Permeabilization Solution Kit (BD Biosciences; 554714). Cells were gated as lymphocytes (FSC-A vs. SSC-A), and single cells (FSC-A vs. FSC-H) as shown in Supplementary Fig. 9. Naive cells were gated as live/single, B220⁺ CD38^Hi FAS^Lo. Plasma cells were gated as live/single, CD138⁺. GC cells were gated as live/single, B220⁺ CD38^Lo FAS^Hi. DZ and LZ cells were gated as CXCR4^hi CD86^lo and CXCR4^lo CD86^hi. Stained cell suspensions were analyzed using a CytoFlex flow cytometer (Beckman Coulter). For RNA-seq, cells were sorted for lack of marker expression (dump⁻: CD4⁻, CD8⁻, GR-1⁻, and F4/80⁻) in addition to expression of GC, LZ and DZ markers, and 30,000 cells were sorted directly into 100 μl Dynabeads mRNA direct kit lysis/binding buffer (Life Technologies;

## Table 1 | Antibodies used in flow cytometry and western blot

| Name | Dilution | Clone | Manufacturer |
|---|---|---|---|
| **Flow cytometry** | | | |
| CD45R/B220-APC-e780 | 1:400 | RA3-6B2 | Invitrogen 47-0452-82 |
| CD38-AF700 | 1:400 | 90 | Invitrogen 56-0381-82 |
| CD95/FAS-PE/Cy7 | 1:400 | Jo2 | BD Biosciences 557653 |
| IgM-APC-e710 | 1:400 | II/41 | eBioscience 46-5790-82 |
| IgG1-FITC | 1:400 | RMG1-1 | Biolegend 406606 |
| IgG1-BV421 | 1:400 | RMG1-1 | Biolegend 406616 |
| IgG2a/b-APC | 1:400 | X-57 | Miltenyi Biotec 130-117-523 |
| IgG3-Biotin | 1:400 | RMG3-1 | Biolegend 406803 |
| IgA-FITC | 1:400 | RMA-1 | Biolegend 11-4204-81 |
| CD138-BV605 | 1:400 | 281-2 | Biolegend 142516 |
| CD86-APC | 1:400 | GL-1 | Biolegend 105012 |
| CXCR4-BV421 | 1:400 | L276F12 | Biolegend 146511 |
| c-Myc | 1:400 | D84C12 | Cell Signaling CTS-5605S |
| CCND3 | 1:400 | SP207 | Abcam ab245734 |
| Streptavidin-APC-e780 | 1:400 | | Invitrogen 47-4317-82 |
| Goat anti-rabbit AF488 | 1:400 | | Abcam ab150077 |
| **Western blot** | | | |
| Beta-actin | 1:1000 | D6A8 | Cell Signaling CTS-84575 |
| DYRK1A | 1:1000 | D30C10 | Cell Signaling CTS-87655 |
| DYRK1A | 1:1000 | 7D10 | Abnova H00001859-M01 |
| MSH6 | 1:1000 | OTI5D1 | Origene TA807929 |
| Thiophosphate ester | 1:5000 | 51-8 | Abcam ab92570 |
| HRP anti-mouse IgG | 1:10,000 | NXA931 | Cytiva NXA931 |
| HRP anti-rabbit | 1:10,000 | NA934 | Cytiva NA934 |

61012) using a FACS ARIA cell sorter (BD), and immediately frozen on dry ice. For NP-specific GC B cells, 10 μg of NP-CGG (Biosearch Technologies) was conjugated to AF488 using the Lightning-Link kit (Abcam; ab236553), according to the manufacturer's instructions and added to the antibody staining mixture at 1:300.

### ELISA
Serum was collected from unimmunized mice, and IgM, IgA, IgG1, and IgG2b titers were determined by ELISA using goat anti-mouse IgM (ab97230), IgA (ab97235), IgG1 (ab97240), and IgG2b (ab97250)–horseradish peroxidase (Abcam) at 1:2500, respectively.

For VSV-specific antibodies in the sera, 96-well plates were coated overnight with inactivated VSV[73] at a concentration of $1 \times 10^6$ PFU/ml. The next day, sera were added at a dilution of 1/50. HRP-conjugated goat anti-mouse IgM, IgG1, IgG2b, and IgG3 from the SBA Clonotyping System-C57BL/6-HRP kit (Southern Biotech; 5300-05B) were used at a 1/500 dilution to detect VSV-specific antibodies. The color intensity of each well was measured using a microwell spectrophotometer read at 450 nm, with a reference filter of 630 nm.

### In vitro CSR and proliferation assays
Spleens were removed, washed in cold PBS, and forced through a 70-μm mesh into PBS containing 2% FCS and 1 mM EDTA to create single-cell suspensions. For the proliferation assay, splenic cells were stained with CellTrace Violet (Invitrogen) according to the manufacturer's instructions. Cells were seeded at $1 \times 10^6$/ml in a 24-well plate and incubated at 37 °C in B-cell medium (RPMI-1640 medium supplemented with 10% FBS, 100 μg/ml penicillin/streptomycin, 50 μg/ml gentamycin, 2 mM glutamine and pyruvate, nonessential amino acids, and 50 μM β-mercaptoethanol). For CSR assay, cells were stimulated with either 50 μg/ml LPS + 50 ng/ml mouse IL-4, or 50 μg/ml LPS for 3 days. For the proliferation assay, cells were stimulated with either

50 μg/ml LPS + 50 ng/ml mouse IL-4, or 50 μg/ml LPS or αIgM for 3 days. CellTrace Violet dilution was assessed by flow cytometry.

### EdU proliferation assay

NP-KLH immunized mice were injected i.v. with 2 mg of the nucleoside analog 5-ethynyl-2′-deoxyuridine EdU (Molecular Probes) in PBS. After 2.5 h, popliteal LNs were stained for the surface antigens B220, CD38, FAS, and CD138, followed by EdU detection using the Click-iT EdU Alexa Fluor 647 Flow Cytometry Assay Kit (Molecular Probes; C10419) according to the manufacturer's instructions. 7AAD (BD Biosciences) was added at 1:50 dilution 5 min before analysis by flow cytometry.

For dual-labeling experiments, NP-KLH immunized mice were injected i.v. with 1 mg of EdU in PBS. After 1 h, the mice were injected i.v. with 2 mg of BrdU (BD Biosciences; 51-2354AK) in PBS. After an additional 3.5 h, popliteal LNs were removed and stained for surface antigens, followed by EdU detection using the Click-iT EdU Alexa Fluor 647 Flow Cytometry Assay Kit (Molecular Probes; C10419) and BrdU detection using the FITC BrdU Flow Kit (BD Biosciences; 51-2354AK) according to the manufacturer's instructions. 7AAD (BD Biosciences) was added at a 1:50 dilution 5 min before analysis by flow cytometry.

### TPLSM image acquisition

A Zeiss LSM 880 upright microscope fitted with a Coherent Chameleon Vision laser was used for imaging experiments. Whole lymph nodes were dissected, and images were acquired with a femtosecond-pulsed two-photon laser tuned to 940 nm. The microscope was fitted with a filter cube containing 565 LPXR to split the emission to a PMT detector (with a 579–631-nm filter for germinal center tdTomato fluorescence). Tile images were acquired as 100–200 μm Z stacks with 5μm intervals between each Z plane. The zoom was set to 1.5, and images were acquired at 512 × 512 x–y resolution. Quantification of the GC area was done using the surface module of Imaris software (Bitplane).

### In vitro kinase assay

Overall, 100 ng of recombinant DYRK1A (ThermoScientific), 5 mg of recombinant MSH6 (Origene), 200 μM ATP-g-S (Abcam), and kinase buffer (40 mM Tris, pH 7.5, 10 mM MgCl₂, 50 mM NaCl) were combined in a 30 μL reaction. Samples were then placed in a Thermocycler at 30 °C and rotated at 1000 rpm for 30 min. To alkylate the proteins, 2.5 mM of p-Nitrobenzyl mesylate (PNBM; Abcam) was added, and the reaction was allowed to proceed for 2 h at room temperature. Afterward, loading dye was added, and samples were heated for 5 min at 95 °C and immediately run for western blotting. Samples were loaded into Biorad Mini-PROTEAN TGX precast gels and run on the Bio-Rad PowerPac HC system. Blots were transferred using the Bio-Rad Trans-Blot Turbo Transfer System. Samples were then blocked in 5% milk in TBST for 1 h at room temperature. Primary antibodies (1:1000) were incubated in 5% milk overnight at 4 °C. Samples were washed three times in 5-min increments with TBST at room temperature before incubation with secondary antibody (1:10,000). Blots were visualized using the Biorad Chemidoc system and analyzed using Image Lab 6.1. All antibodies used are listed in Table 1.

### Retrovirus production and transduction

Platinum-E cells (Cell Biolabs) were grown to 70–80% confluence in 10-cm dishes and transfected with 10 μg of pMSCV:IRES:EGFP MSH6 WT or MSH6 T326A plasmid (VectorBuilder) diluted in 1 ml Opti-MEM and 40 μl Turbofect (ThermoFisher). Retrovirus-containing supernatants were collected 48-72 h after transfection. Fresh virus was concentrated by adding 1/3 viral volume of Retro-X concentrator (Takara Bio), overnight incubation at 4 °C, and centrifugation at 1500×g for 45 min at 4 °C. Splenic B cells from WT mice were column purified using the CD43 (Ly-48) microbead kit (Milteny Biotec; 130-049-801), and 2 × 10⁶ cells were stimulated overnight at a final concentration of 2.5 μg/ml (1:400 dilution) αCD180 (Biolegend, clone RP/14) diluted in B-cell media. The next day, stimulated B cells were washed and transduced by adding concentrated virus resuspended in B-cell media and Polybrene (Sigma) at a final concentration of 10 μg/ml. Spinoculation was performed at 1200×g for 90 min at 32 °C. Cells were then incubated for 3 h at 37 °C, followed by a second spinoculation. Transduced cells were then incubated at 37 °C for 48 h, washed four times, and stimulated with 20 μg/ml LPS and 40 ng/ml mouse IL-4 for 3 days.

### Western blot analysis

Naive splenic B cells were isolated using anti-CD43 magnetic beads (Miltenyi Biotec; 130-049-801). Purified B cells were kept unstimulated, or stimulated with 10 μg/m LPS for 3 days. The cells were then lysed in radioimmunoprecipitation assay buffer (10 mM Tris-HCl, 1 mM EDTA, 0.5 mM EGTA, 1% Triton-100 140 mM NaCl, 0.1% deoxycholate, 0.1% SDS). Lysates were centrifuged for 15 min at 4 °C. Cleared lysates were boiled with sample buffer for 5 min, separated by SDS-PAGE (Bio-Rad), and transferred to nitrocellulose membranes. Blots were blocked with 5% skim milk in TBST and 0.05% Tween-20 for 1 h at room temperature, and incubated with primary antibody diluted 1:1000 overnight at 4 °C. Horseradish peroxidase-conjugated donkey anti-rabbit secondary antibody and ECL Reagent (Biological Industries) were used for detection. All antibodies are listed in Table 1.

### Single-cell IgH sequencing

Popliteal LNs from immunized mice or Peyer's patches from aged mice (1 year old) were harvested and processed for flow cytometry analysis. Cell suspensions were stained for dump⁻ (CD4, CD8, GR-1, F4/80) and B220, CD38, FAS and IgG1 (BioLegend) expression. Cell sorting was performed using a FACS Aria cell sorter (BD Bioscience). GC cells were gated as live/single, B220⁺ CD38^Lo FAS^Hi. GC-derived IgG1 B cells were sorted into 96-well plates containing lysis buffer (PBS with 3 U/μl RNAsin, 10 mM dithiothreitol). cDNA was purified using random primers (NEB), as previously described[74]. Nested PCR was used to amplify a segment of Igγ1 heavy chains using the outer constant primers (5′-GGAAGGTGTGCACACCGCTGGAC-3′) together with a mix of primers for the variable regions[74], followed by a second reaction with the inner constant (5′-GCTCAGGGAAATAGCCCTTGAC-3′) and variable primers (5′- GGGAATTCGAGGTGCAGCTGCAGGAGTCTGG-3′). Amplification conditions were as follows: 98 °C for 30 s, 30 cycles of [98 °C for 30 s, 50 °C for 30 s, and 72 °C for 30 s] (reaction I), or 40 cycles of [98 °C for 30 s, 55 °C for 30 s, and 72 °C for 30 s] (reaction II), followed by 72 °C for 2 min. PCR products were sequenced by Sanger sequencing. Sequences were aligned to the IMGT mouse heavy chain gene database (downloaded Dec. 2019)[75] using NCBI IgBlast v1.17.0[76], and processed using Change-O v 1.2.0[77]. Downstream analysis of clustering, mutational load, and diversity was performed using Change-O v1.2.0[77], Alakazam v1.2.0[77], SHazaM v1.1.0[77], and custom scripts within the R v4.1.0 statistical computing environment. Clonal inference of the V(D)J sequences was based on identical IGHV and IGHJ gene annotations, and the length of the junction region. Based on these inferences, full germline sequences and phylogenetic trees were constructed for each clone. The analysis of the mutations and diversity was deduced from the phylogenetic tree of each clone using a custom R v4.1.0 script.

### Mutation analysis of the upstream region of the core Sμ

B cells were stimulated with 50 μg/ml LPS + 50 ng/ml mouse IL-4 for 3 days, and genomic DNA was isolated using the DNeasy blood and tissue kit (Qiagen; 69504). Next, 10 ng of DNA was used to amplify the upstream of the core S region using Platinum SuperFi II PCR Master mix (Invitrogen) with the following primers: 5′-AATGGATACCTCAGTGGTTTTTAATGGTGGGGTTTA-3′ and 5′-GCGGCCCGGCTCATTCCAGTTCATTACAG-3′, and PCR conditions: 98 °C 1 min, 35 cycles of 98 °C for 20 s, 50 °C for 30 s, and 72 °C for 1 min, followed by a final incubation at 72 °C for 10 min. Amplified PCR products (560 bp) were cloned using the Zero Blunt TOPO PCR cloning kit (Invitrogen; 45-

0159). Plasmid DNA from multiple colonies was prepared using the Presto miniplasmid kit (Genaid; PDH300). T7 primers were used to sequence the PCR inserts, and sequence alignments were constructed by comparing the PCR sequence with the germline sequence.

## RNA sequencing

Popliteal LNs from mice immunized 7 days previously were harvested and sorted for GC DZ and LZ B cells based on the following staining: dump − (CD4, CD8, GR-1, F4/80) and B220 + CD38 − FAS + , followed by CXCR4 + CD86 − or CXCR4 − CD86 + , representing DZ or LZ cells, respectively. For gene expression analysis, $3 \times 10^4$ cells from each population were sorted into 100 µl lysis/binding buffer, and mRNA was captured using the Dynabeads mRNA direct kit according to the manufacturer's instructions (Life Technologies; 61012). A bulk adaptation of the massively parallel single-cell RNA sequencing protocol (MARS-seq) was used as previously described[78,79] to generate RNA-seq libraries for transcriptomic analysis. Alignment and differential expression analysis was performed using the UTAP pipeline v1.10.2[80]. Reads were trimmed using Cutadapt and mapped to the Mus musculus genome (UCSC mm10) using STAR[81] v2.4.2a with default parameters. The pipeline quantifies the genes annotated in RefSeq (extended by 1000 bases toward the 5′ edge and 100 bases in the 3′ direction). Htseq-count[82] (union mode) was used for counting sequenced reads. Expression analysis was based on genes with a minimum of five UMI-corrected reads in at least one sample. Normalization of the counts and differential expression analysis was performed using DESeq2[83]. Raw $P$ values were adjusted for multiple testing using the procedure of Benjamini and Hochberg. Differentially expressed genes were visualized using the EnhancedVolcano R package. The threshold for significant differential expression was $\log_2 FC > 0.58$ or $< -0.58$, $P < 0.05$. GSEA was performed using GSEA 4.1 with the GSEA preranked tool[84]. Gene names were converted to human gene symbols, and run with default parameters. The Molecular Signature Database hallmark gene sets were used to perform pathway enrichment analysis using a hypergeometric distribution, and limiting the output to the top 100 gene sets. The Metascape v3.5.20230101 tool was used to define unique pathways that were significantly modified using the threshold mentioned above[38].

## Sample preparation for proteomics analysis

Splenic B cells were column purified using the CD43 (Ly-48) microbead kit (Milteny Biotec; 130-049-801), and $8 \times 10^6$ cells were stimulated with 10 µg/ml LPS or 10 µg/ml αIgM or 50 µg/ml LPS + 50 ng/ml mouse IL-4 for 3 days. Cells were washed twice with cold PBS, and resuspended in 80 µl lysis buffer (50 mM Tris pH 7.6, 5% SDS and 1% phosphatase inhibitor cocktails 2 and 3 (Sigma)). Lysates were incubated at 95 °C for 5 min, and sonicated for six cycles of 30 s (Bioruptor Pico, Diagenode, USA). Protein concentration was measured using the BCA assay (Thermo Scientific, USA), and a total of 120 µg protein was reduced with 5 mM dithiothreitol (Sigma) and alkylated with 10 mM iodoacetamide (Sigma) in the dark. Each sample was loaded onto S-Trap minicolumns (Protifi, USA) according to the manufacturer's instructions. In brief, after loading, samples were washed with 90:10% methanol/50 mM ammonium bicarbonate. Samples were then digested with trypsin (1:50 trypsin/protein) for 1.5 h at 47 °C. The digested peptides were eluted using 50 mM ammonium bicarbonate; trypsin was added to this fraction and incubated overnight at 37 °C. Two more elutions were performed using 0.2% formic acid and 0.2% formic acid in 50% acetonitrile. The three eluted fractions were pooled, and vacuum-centrifuged to dry. Samples were maintained at −20 °C until analysis[85].

## Immobilized metal affinity chromatography

For chromatography, 115 µg of each sample was subjected to phosphopeptide enrichment. Enrichment was performed on a Bravo robot (Agilent Technologies) using AssayMAP Fe(III)-NTA, 5 µl cartridges (Agilent Technologies), according to the manufacturer's instructions. In brief, cartridges were primed and equilibrated with 50 µl of buffer A (99.9% ACN/0.1% TFA) and 100 µl of buffer C (80% ACN/19.9% H2O/ 0.1% TFA), followed by sample loading in 100 µl of buffer C at 5 µl/min. Phosphopeptides were eluted with 120 µl of buffer B (99% H2O/1% NH3) at 5 µl/min. Next, 3 µl of formic acid was added to each sample for acidification. Prior to LC−MS analysis, all samples were concentrated to a volume of 15 µl[85].

## Liquid chromatography

ULC/MS-grade solvents were used for all chromatographic steps. Each sample was loaded using a split-less nano-Ultra Performance Liquid Chromatography column (10 kpsi nanoAcquity; Waters, Milford, MA, USA). The mobile phases were: (A) $H_2O + 0.1\%$ formic acid, and (B) acetonitrile + 0.1% formic acid. Desalting of the samples was performed inline using a reversed-phase Symmetry C18 trapping column (180 µm internal diameter, 20 mm length, 5-µm particle size; Waters). The peptides were then separated using a T3 HSS nano-column (75 µm internal diameter, 250 mm length, 1.8-µm particle size; Waters) at 0.35 µL/min. Peptides were eluted from the column into the mass spectrometer using the following gradient: 4% to 33% B (for the global proteomics) or 20% B (for the phosphoproteomics) for 155 min, then to 90% B for 5 min, maintained at 90% for 5 min, and then returned to initial conditions[85].

## Mass spectrometry

The nanoUPLC was coupled inline through a nanoESI emitter (10-µm tip; New Objective; Woburn, MA, USA) to a quadrupole orbitrap mass spectrometer (Q Exactive HF, Thermo Scientific) using a FlexIon nanospray apparatus (Proxeon). Data were acquired in data-dependent acquisition (DDA) mode, using a Top10 method. MS1 resolution was set to 120,000 (at 200 $m/z$), mass range of 375–1650 $m/z$, AGC of 3e6, and maximum injection time was set to 60 msec. MS2 resolution was set to 15,000, quadrupole isolation 1.7 $m/z$, AGC of 1e5, dynamic exclusion of 45 sec, and maximum injection time of 60 msec for the global proteomics and 150 msec for the phosphoproteomics[85].

## Proteomics data processing

Raw data were processed with MaxQuant v1.6.6.0[86,87]. The data were searched with the Andromeda search engine against the Uniprot human proteome database appended with common lab protein contaminants and the following modifications: Carbamidomethylation of C was noted as a fixed modification, oxidation of M and protein N-terminal acetylation as variable ones. For the phospho-sites analysis, phosphorylation of S, T and Y were added, as well. The remaining parameters were kept at default values, except for the following: min. peptide length was set to 6, label and LFQ min. ratio count were set to 1, match between runs and iBAQ calculation were enabled, and the protein quantification was done on the basis of unique peptides only. The LFQ intensities (Label-Free Quantification) were extracted and used for further calculations using Perseus v1.6.2.3[88]. Decoy hits were filtered out, as well as proteins that were identified on the basis of a modified peptide only. The data were further filtered to include only proteins with at least three valid values in at least one of the groups. Protein expression imputation was done with a random low range normal distribution. A Student's $t$ test, after logarithmic transformation, was used to identify significant differences across the biological replica. Fold changes were calculated based on the ratio of geometric means of the different compared groups. Phospho analysis was done using the phospho-sites table generated by Maxquant. The data were filtered as in the global analysis. The intensities were normalized by subtracting the median, and missing values were imputed by a low constant. Statistics were performed similarly as for protein expression. Significant changes in protein abundance and phosphorylation levels were visualized using the EnhancedVolcano R package (v4.2.0). A site

was considered as differentially phosphorylated if its phosphorylation level significantly changed compared to WT (FC > 1.5 or < −1.5, $P < 0.05$). Protein sites were excluded when the protein level was significantly changed in the same direction as the phosphorylation level (FC > 1.25 or < −1.25, $P < 0.1$). The Metascape tool was used to define unique pathways that were significantly modified using the threshold mentioned above[38].

## Statistical analysis

Statistical significance was determined with GraphPad Prism Version 9.0 using the tests indicated in each figure.

## Reporting summary

Further information on research design is available in the Nature Portfolio Reporting Summary linked to this article.

## Data availability

The mass spectrometry proteomics data generated in this study have been deposited in the ProteomeXchange Consortium via the PRIDE partner repository, with the dataset identifier PXD034156. Proteomic data processing was done using the Uniprot human proteome database (https://www.uniprot.org/proteomes/UP000005640). The single-cell RNA sequencing data generated in this study were deposited in the NCBI's Gene Expression Omnibus database, with the dataset identifier GSE206146. IgH sequences were aligned to the IMGT mouse heavy chain gene database using NCBI IgBlast (https://www.ncbi.nlm.nih.gov/igblast/). Gene names were aligned to the Molecular Signature Database hallmark gene sets (http://www.gsea-msigdb.org/gsea/msigdb/human/collections.jsp#H). The remaining data are available within the paper, Supplementary Information, or Source Data file. Source data are provided with this paper.

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

## Acknowledgements

Z.S. is supported by the European Research Council (ERC) grant No. 101001613, by the Morris Kahn Institute for Human Immunology, Human Frontiers of Science Program (CDA-00023/2016), Azrieli Foundation, Israel Science Foundation (ISF, 1090/18). Z.S. is a member of the European Molecular Biology Organization (EMBO) Young Investigator Program. Z.S. is supported by grants from The Benoziyo Endowment Fund for the Advancement of Science, The Sir Charles Clore Research Prize, Comisaroff Family Trust, Irma & Jacques Ber-Lehmsdorf Foundation, Gerald O. Mann Charitable Foundation and David M. Polen Charitable Trust and Elie Hirschfeld and Dr. Sarah Schlesinger. E.H. received support from the ASH Medical Student Physician-Scientist Award.

## Author contributions

L.S.B. designed and conducted the experiments, performed data analysis, and wrote the manuscript. E.H. performed kinase assays. A.P. analyzed single-cell immunoglobulin-sequencing data. H.H. analyzed RNA-seq. data. M.Kuk. generated and provided VSV virus. P.D.L. performed VSV-specific ELISA. A.G. conducted some FACS experiments. N.G. performed western blot experiment. M.Kup. performed and analyzed proteomics experiments. B.H.Y. provided protocols for retrovirus experiments. M.I. supervised VSV production and VSV-specific ELISA experiments. G.Y. supervised immunoglobulin-sequencing analysis. J.D.C. supervised kinase assays. Z.S. designed experiments, supervised the study, and wrote the manuscript.

## Competing interests

The authors declare no competing interests.
