## [Peer Review File · Nature Communications]

B cell class switch recombination is regulated by DYRK1A through MSH6 phosphorylationREVIEWER COMMENTS

Reviewer #1 (Remarks to the Author):

The authors show that protein kinase DYRK1A plays an essential role in immunoglobulin class switch recombination (CSR) in mice. This finding is highly relevant to the field, novel and properly addressed by using sophisticated mouse models and analyses. To elucidate the mechanism involved the authors perform (phospho)proteomics and found that DYRK1A acts by phosphorylating MSH6, a mismatch repair protein known to be involved in CSR. Although the data indeed points towards involvement of MSH6, this part is less convincing.

Major points:

In general, the data concerning in vivo experiments shown in Figures 3-6 are convincing and clear. However, there are 2 major concerns related to the in vitro experiments shown in Figures 1 and 2.

- CSR is the main topic of this manuscript. It is therefore remarkable that the authors used very poor CSR-inducing conditions for their in vitro cultures, namely LPS+IL-4, where only 5% of control B cells induce CSR (Figure 1D). Although it has indeed been published decades ago that IL-4 addition promotes CSR of naive B cells (eg. Janas et al. *J Immunol.* 1999 Oct 15;163(8):4192-8), much better in vitro cultures conditions that induce CSR have been described more recently. For example, Nojima et al. showed CSR of naive murine B cells in their culture model in 77% of cells (30% IgG1 and 47% IgE) (*Nat Commun.* 2011 Sep 6;2:465). I suggest to repeat the experiments shown in Figure 1D under these, or similar, conditions to improve the readout and the validity of the findings related to DYRK1A.
- The experiments in Figure 2 to identify targets of DYRK1A are performed in cultures with either LPS or with anti-IgM. These settings are not/only poorly inducing CSR in stimulated B cells. It is not clear to me why the authors first show that DYRK1A is involved in CSR and subsequently choose these culture conditions to find relevant phosphorylation targets. I suggest to repeat these experiments in the presence of IL-4 and/or CD40L, or using culture conditions as described above by Nojima et.

Minor points:

- The semi-last sentence in the results section related to figure 1 (of note,....with LPS) does not make sense or is unfinished. This issue should be addressed properly since CSR has previously been shown to be division-linked (Hodgkin et al. *J Exp Med.* 1996 Jul 1;184(1):277-81 and Rush et al. *Proc Natl Acad Sci U S A.* 2005 Sep 13;102(37):13242-7).
- In figure 4D the authors show enlarged germinal centers in the absence of DYRK1A. The authors, as well as previous publications, indeed show that DYRK1A regulates cell division. However, it is also known that anti-apoptotic BCL-2 family proteins control the size of germinal centers (eg. Vikstrom et al. *Science* 2010 Nov 19;330(6007):1095-9). Since it is not examined whether DYRK1A interferes with expression of BCL-2 family proteins, the possibility should not be excluded and should be addressed textually.

Reviewer #2 (Remarks to the Author):

This paper examines the role of DYRK1A in B cell biology. They demonstrate that DYRK1A is involved in regulating class switch recombination, at least in part by regulating MSH6, and in restraining B cell proliferation. This expands on what was previously known about the role of DYRK1A in B cell development. While this is interesting work there were some areas that needed further clarification to ensure that the data fully supports the conclusions before publication as detailed below.

The argument for a lack of antigen-specific clonal expansion is not entirely convincing. Perhaps this is a language issue but given the increased GC it seems more likely that there is increased Ag non-specific clonal expansion which masks the Ag-specific clonal expansion rather than the lack of Ag-specific clonal expansion itself. Indeed the normal affinity maturation when they focus on IGHV11-72 clones, the increased GC and the increased proliferation would suggest that the problem is less a failure of clonal expansion and more a problem of inappropriate clonal expansion. This should be clarified by:

- providing data on the size of the GC at day 21 for WT and cKO
- Examining whether there is a change in absolute # of Ag-specific cells not just in the relative proportions?
- Looking at the proliferation of Ag-specific cells, not just total GC cells.

The proliferation defect is interesting but it is unclear how this leads to the expansion of the LZ – what is the rate of proliferation in the DZ v the LZ. Can they show the data in Fig 6E and F split for LZ and DZ?

DYRK1A has previously been shown to phosphorylate Myc and yet the expression of Myc targets are decreased in the DYRK1A cKO mice – (Fig 6F). This seems at odds with their discussion which suggests “We propose that enhancing Myc expression and proliferation of all the cells in the GC will raise the bar for the selection of specific clones for expansion” Did they examine Myc expression in the GC? Does Myc come up in their proteomics/phosphoproteomics screens? Their conclusions on the mechanisms of altering proliferation could also be strengthened by linking their transcriptomic, proteomic and proliferation data.

Minor points

In the in vitro culture and in the BM PCs there are a large number of IgM- IgG1- cells. Would be good to show staining for other isotypes to demonstrate that this defect also affects other isotypes. In particular, IgG3 switching can be seen with LPS alone and would fit well with their demonstration that MSH6 phosphorylation is associated with decreased phosphorylation of MSH6 in response to LPS alone.

Page 5 – They say that the lack of effective proliferation could not account for the diminished CSR but don't show the proliferation for cells cultured with LPS+ IL-4 – please include this data

“The GC reaction is the major source for class-switched antibody-secreting cells.” Would be more correct here to say plasma cells rather than antibody-secreting cells as there are a lot of short-lived extra follicular class-switched antibody secreting cells.

Would be good to see the list of other proteins that were differentially regulated – can they include this in supplementary data.

VSV model – Are they able to measure the VSV specific IgM and switched isotype Ab titres so this can be correlated with survival

What is the explanation for the failure to observe increased GCs in the CD23-cre mice? Is this due to the decreased switching? A brief explanation of this would be helpful in the discussion.

The discussion states “Whereas CSR did not take place in Dyrk1a-deficient B cells in vitro, a small IgG1+ population was detected in vivo” The data in Fig 1 shows a small number of switched cells not a complete absence. Indeed, there is about a 5-fold decrease in the % of switched cells in vitro which is a similar fold decrease to what is seen in vivo suggesting that this is relative to the total amount of switching that is going on. Therefore, would be better to rephrase this sentence.

Reviewer #3 (Remarks to the Author):

The manuscript by Stoler-Barak describes an essential role for the protein kinase DYRK1A in B-cell immune responses, through regulation of class switch recombination (CSR) and clonal expansion of antigen-specific B-cells within germinal centres (GCs). The DYRK family of protein kinases have been

extensively studied in the context of cancer biology and hold great therapeutic potential; nevertheless, and despite their function as master regulators of cell proliferation, their roles in B-cell activation has so far remained unexplored. The work is original and well-presented, using appropriate methodology to address relevant questions for the biology of B-cell immune responses and antibody maturation. But the manuscript could be strengthened in providing stronger evidence supporting their main claim, that CSR is regulated by DYRK1A through MSH6 phosphorylation.

I have three main points to be addressed by the authors:

1. Experiments in Fig. 2E-G supporting a role for DYRK1A-mediated MSH6 phosphorylation promoting CSR should be further substantiated to establish this as the main mechanism by which DYRK1A impacts on CSR.
 - a. Can a MSH6 phosphomimic mutant bypass the CSR defects observed in DYRK1A deficient B-cells?
 - b. What is the effect of MSH6 phosphorylation on its mismatch repair activity? S-region mutation analyses in B-cells deficient for MSH6 show fewer A:T substitutions; a similar analysis in DYRK1A deficient B-cells or, ideally, B-cells expressing MSH6-T326A would help clarify how MSH6 phosphorylation impacts on mismatch repair during CSR.
2. The analysis of nucleotide base substitutions in GC B-cells suggests that the main role for DYRK1A in the GC is not the regulation of SHM via MSH6. This raises questions as to why DYRK1A-mediated MSH6 phosphorylation is specifically needed for CSR. The authors
 - a. Is MSH6 phosphorylation by DYRK1A specifically required for MSH6 targeting at switch recombination sites?
 - b. Is MSH6 protein, through DYRK1A phosphorylation, involved in the generation and/or resolution of DSBs at switch recombination sites?
3. It is difficult to reconcile that in AID-Cre DYRK1A fl/fl mice, an enlarged GC phenotype is concomitant with a diminished dark-zone compartment, where most GC B-cell proliferation occurs.
 - a. The authors suggests that an increase in GC size in AID-Cre DYRK1A fl/fl mice is the result of enhanced proliferation in the light-zone. This should be directly tested using dual-labelling with EdU and BrdU, to determine cell-cycle dynamics in both dark-zone and light-zone compartments.
 - b. DYRK1A has been shown to control pre-B cell proliferation through destabilization of CyclinD3 protein levels upon phosphorylation. CyclinD3 drives 'inertial' cell cycle in dark-zone B-cells through sustained E2F expression – do the authors find evidence for changes in CyclinD3 protein levels in DYRK1A deficient GC B-cells?

Reviewer #1 (Remarks to the Author):

The authors show that protein kinase DYRK1A plays an essential role in immunoglobulin class switch recombination (CSR) in mice. This finding is highly relevant to the field, novel and properly addressed by using sophisticated mouse models and analyses. To elucidate the mechanism involved the authors perform (phospho)proteomics and found that DYRK1A acts by phosphorylating MSH6, a mismatch repair protein known to be involved in CSR. Although the data indeed points towards involvement of MSH6, this part is less convincing.

Major points:

In general, the data concerning in vivo experiments shown in Figures 3-6 are convincing and clear. However, there are 2 major concerns related to the in vitro experiments shown in Figures 1 and 2.

We appreciate the referee's positive evaluation of our study, and are pleased that he/she found the results of the experiments to be convincing. We address the referee's comments below:

- CSR is the main topic of this manuscript. It is therefore remarkable that the authors used very poor CSR-inducing conditions for their in vitro cultures, namely LPS+IL-4, where only 5% of control B cells induce CSR (Figure 1D). Although it has indeed been published decades ago that IL-4 addition promotes CSR of naive B cells (eg. Janas et al. J Immunol. 1999 Oct 15;163(8):4192-8), much better in vitro cultures conditions that induce CSR have been described more recently. For example, Nojima et al. showed CSR of naive murine B cells in their culture model in 77% of cells (30% IgG1 and 47% IgE) (Nat Commun. 2011 Sep 6;2:465). I suggest to repeat the experiments shown in Figure 1D under these, or similar, conditions to improve the readout and the validity of the findings related to DYRK1A.

Indeed the efficiency of CSR in our experiment was not optimal, yet the differences between *Dyrk1a*-deficient and WT B cells were very clear, and statistically significant. As suggested by the referee, we repeated the experiments wherein we stimulated the B cells with a higher dose of LPS and IL-4, which resulted in about 4-fold more IgG1-expressing B cells compared to the original experiments (20% of total B cells, on average). Since this stimulation of B cells with LPS and IL-4 is considered the gold standard of CSR, we refrained from using the iGB (in vitro-generated germinal center) protocol as described in Nojima et al, which involves additional factors that are not purely related to CSR. The new data are shown in Fig. 1d.

- The experiments in Figure 2 to identify targets of DYRK1A are performed in cultures with either LPS or with anti-IgM. These settings are not/only poorly inducing CSR in stimulated B cells. It is not clear to me why the authors first show that DYRK1A is involved in CSR and subsequently choose these culture conditions to find relevant phosphorylation targets. I suggest to repeat these experiments in the presence of IL-4 and/or CD40L, or using culture conditions as described above by Nojima et.

We agree with the reviewer that CSR proteomic analysis using LPS+IL4 would have been ideal for this type of experiment. Of note, in the same figure, we demonstrated that MSH6 is a target of DYRK1A using recombinant proteins in a kinase assay. Thus, our original conclusions remain valid regardless of the initial screening method. We addressed the referee's comments in two ways:

1. We took the referee's advice and repeated the proteomic analyses using LPS+IL4. We found a reduction in MSH6 phosphorylation ($p=0.06$ statistical significance) and changes in DNA damage and repair pathways. Because of time limitations, the need to perform many experiments with a limited number of mice, and ethical constraints, we used fewer mice in this experiment than in the original one (LPS without IL-4), which led to a slightly less-significant result. These results are shown in Supplementary Fig. 2.

Supplementary Fig. 2

2. LPS without IL-4 induces CSR to IgG3 and IgG2a/b, which is more relevant to our initial proteomic analysis. Using LPS stimulation, we found in *Dyrk1a*-deficient B cells, a significant defect in CSR to these isotypes as well. These results are now part of Fig. 1d.

Minor points:

- The semi-last sentence in the results section related to figure 1 (of note,....with LPS) does not make sense or is unfinished. This issue should be addressed properly since CSR has previously been shown to be division-linked (Hodgkin et al. J Exp Med. 1996 Jul 1;184(1):277-81 and Rush et al. Proc Natl Acad Sci U S A. 2005 Sep 13;102(37):13242-7).

We corrected this sentence in the revised manuscript and added the relevant references:

“It was previously shown that CSR is linked to cell division^{36,37}; however, since *Dyrk1a*-deficient B cells stimulated with LPS+IL-4 showed comparable rather than diminished proliferation in-vitro, we conclude that reduced cell division does not explain their CSR defect.”

- In figure 4D the authors show enlarged germinal centers in the absence of DYRK1A. The authors, as well as previous publications, indeed show that DYRK1A regulates cell division. However, it is also known that anti-apoptotic BCL-2 family proteins control the size of germinal centers (eg. Vikstrom et al. Science 2010 Nov 19;330(6007):1095-9). Since it is not examined whether DYRK1A interferes with expression of BCL-2 family proteins, the possibility should not be excluded and should be addressed textually.

The following text was added to the Discussion section:

“BCL-2 reduces B cell apoptosis in the GC, and its enhanced expression is associated with increased GC size^{69,70}. Although we did not detect changes in the expression of BCL2 family members by RNA-seq, a role for DYRK1A in regulating apoptosis in the GC cannot be excluded.”

Reviewer #2 (Remarks to the Author):

This paper examines the role of DYRK1A in B cell biology. They demonstrate that DYRK1A is involved in regulating class switch recombination, at least in part by regulating MSH6, and in restraining B cell proliferation. This expands on what was previously known about the role of DYRK1A in B cell development. While this is interesting work there were some areas that needed further clarification to ensure that the data fully supports the conclusions before publication as detailed below.

We thank the reviewer for her/his positivity and helpful comments.

The argument for a lack of antigen-specific clonal expansion is not entirely convincing. Perhaps this is a language issue but given the increased GC it seems more likely that there is increased Ag non-specific clonal expansion which masks the Ag-specific clonal expansion rather than the lack of Ag-specific clonal expansion itself. Indeed the normal affinity maturation when they focus on IGHV11-72 clones, the increased GC and the increased proliferation would suggest that the problem is less a failure of clonal expansion and more a problem of inappropriate clonal expansion.

We carefully examined these issues experimentally. We believe that the experiments described below rule out “Ag non-specific clonal expansion which masks the Ag-specific clonal expansion”.

This should be clarified by:

- providing data on the size of the GC at day 21 for WT and cKO

We examined the size of the GC over time, as the reviewer requested, and found that initially, the GCs are larger, but then become smaller over time, suggesting that *Dyrk1a*-deficiency has divergent effects on early and late stages of the immune response. These results are now part of supplementary Fig. 6.

Supplementary Fig. 6

- Examining whether there is a change in absolute # of Ag-specific cells not just in the relative proportions? Looking at the proliferation of Ag-specific cells, not just total GC cells.

We quantified the number of NP-specific B cells using an antigen labeled with Alexa-488, and found a significant increase in their absolute number on day 7 after the immunization. In addition, the number of antigen-specific cells that undergo proliferation was increased. We conclude that DYRK1A plays a major role in restricting the proliferation of antigen-specific cells at the early GC stages. These data are shown in supplementary Fig. 5.

Supplementary Fig. 5

The proliferation defect is interesting but it is unclear how this leads to the expansion of the LZ – what is the rate of proliferation in the DZ v the LZ. Can they show the data in Fig 6E and F split for LZ and DZ?

Indeed, we raised the possibility that *Dyrk1a*-deficient B cells proliferate more in the LZ. We now examined B cell proliferation in the two GC compartments, and found somewhat enhanced proliferation in the DZ. We modified our initial prediction in the Discussion section, and the new data are now presented in Fig. 6. We also examined B cell proliferation and S-phase entry using EdU and BrdU labeling, as suggested by Reviewer 3. We found enhanced B cell proliferation in the DZ, but not in the LZ (Supplementary Fig. 8). We would like to clarify that DZ size is not substantially diminished, but rather is slightly smaller compared to that of the control mice; however, the GC size is significantly larger in *AID.Cre.Dyrk1a^{fl/fl}* mice. Indeed, we found a 4-fold increase in the number of antigen-specific B cells in the GCs of the *AID.Cre.Dyrk1a^{fl/fl}* mice compared to control. Together, we suggest that the GC size in *AID.Cre.Dyrk1a^{fl/fl}* mice is larger, most likely due to enhanced proliferation that occurs in the DZ.

“The rate of B cell proliferation in the DZ of *AID.Cre.Dyrk1a^{fl/fl}* mice was higher compared to control animals, suggesting that a defect in the pathways restraining the cell cycle in this zone by DYRK1A can explain the increase in global GC size.”

Fig. 6

DYRK1A has previously been shown to phosphorylate Myc and yet the expression of Myc targets are decreased in the DYRK1A cKO mice – (Fig 6F). This seems at odds with their discussion which suggests “We propose that enhancing Myc expression and proliferation of all the cells in the GC will raise the bar for the selection of specific clones for expansion” Did they examine Myc expression in the GC? Does Myc come up in their proteomics/phosphoproteomics screens? Their conclusions on the mechanisms of altering proliferation could also be strengthened by linking their transcriptomic, proteomic and proliferation data.

Based on the referee’s comment, we re-examined our data sets. c-Myc protein was not detected in the global proteomic data, and we did not find changes in *Myc* expression in the RNA-seq data. We now mention these observations in the text. Furthermore, using flow cytometry, we did not find changes in c-Myc and Cyclin D3 protein levels in *Dyrk1a*-deficient B cells (supplementary Fig. 1).

“Using the Metascape database for the analysis of datasets at the systems level³⁸, we found that DYRK1A is involved in DNA damage and repair pathways and in cell cycle progression. However, the proliferation-related proteins c-Myc and Cyclin D3 were not detected in our proteomic analysis (Fig. 2a,b, source data file).”

Supplementary Fig. 1

Minor points

In the in vitro culture and in the BM PCs there are a large number of IgM- IgG1- cells. Would be good to show staining for other isotypes to demonstrate that this defect also affects other isotypes.

We found IgA⁺ plasma cells in the BM, which represent the IgM⁻ IgG⁻ BM PCs (Supplementary Fig. 3).

Supplementary Fig. 3

In particular, IgG3 switching can be seen with LPS alone and would fit well with their demonstration that MSH6 phosphorylation is associated with decreased phosphorylation of MSH6 in response to LPS alone.

We tested the hypothesis suggested by the referee, and found a significant reduction in CSR to IgG3 and IgG2a/b in *Dyrk1a*-deficient B cells (Fig. 1d).

Page 5 – They say that the lack of effective proliferation could not account for the diminished CSR but don't show the proliferation for cells cultured with LPS+ IL-4 – please include this data

We repeated the experiment using LPS+IL4 and did not find a defect in *Dyrk1A*-deficient B cell proliferation. The data are now shown in Fig. 1c.

“The GC reaction is the major source for class-switched antibody-secreting cells.” Would be more correct here to say plasma cells rather than antibody-secreting cells as there are a lot of short-lived extra follicular class-switched antibody secreting cells.

Corrected in the text.

Would be good to see the list of other proteins that were differentially regulated – can they include this in supplementary data.

We now include the complete data set in the source data file.

VSV model – Are they able to measure the VSV specific IgM and switched isotype Ab titres so this can be correlated with survival.

We collected serum from the surviving mice at the end of this experiment (21 days following infection). Using ELISA, we found VSV-specific IgG1, IgG2b and IgG3, suggesting that a parallel pathway overcame the defect in CSR and allowed the mice to survive. Note that, consistent with our ethics protocol and animal handling permissions, we refrained from taking serum samples from the dying mice that lost significant weight, since these animals were suffering. The results are shown in supplementary Fig. 4.

Supplementary Fig. 4

What is the explanation for the failure to observe increased GCs in the CD23-cre mice? Is this due to the decreased switching? A brief explanation of this would be helpful in the discussion.

Indeed we did not detect an increase in GC size in the CD23.Cre model as opposed to AID.Cre mouse models. We suggest the following explanation in the Discussion:

“RNA-seq, a role for DYRK1A in regulating apoptosis in the GC cannot be excluded. Unexpectedly, the GC size in the AID.Cre model was increased, whereas this effect was not observed in the CD23.Cre mouse model. GC formation and maintenance depend on antigen retention on follicular dendritic cells as immune complexes for long periods⁷¹. The reduction in IgG antibodies in the CD23.Cre mouse model may affect the ability of *Dyrk1a*-deficient B cells to seed large GCs, a defect that does not occur in the AID.Cre mouse model.”

The discussion states “Whereas CSR did not take place in *Dyrk1a*-deficient B cells in vitro, a small IgG1+ population was detected in vivo” The data in Fig 1 shows a small number of switched cells not a complete absence. Indeed, there is about a 5-fold decrease in the % of switched cells in vitro which is a similar fold decrease to what is seen in vivo suggesting that this is relative to the total amount of switching that is going on. Therefore, would be better to rephrase this sentence.

We fixed this sentence to more accurately reflect our findings:

“Our results show that MSH6 is a target for DYRK1A, yet a small *Dyrk1a*-deficient, IgG1+ - expressing B cell population was detected in our experiments.”

Reviewer #3 (Remarks to the Author):

The manuscript by Stoler-Barak describes an essential role for the protein kinase DYRK1A in B-cell immune responses, through regulation of class switch recombination (CSR) and clonal expansion of antigen-specific B-cells within germinal centres (GCs). The DYRK family of protein kinases have been extensively studied in the context of cancer biology and hold great therapeutic potential; nevertheless, and despite their function as master regulators of cell proliferation, their roles in B-cell activation has so far remained unexplored. The work is original and well-presented, using appropriate methodology to address relevant questions for the biology of B-cell immune responses and antibody maturation. But the manuscript could be strengthened in providing stronger evidence supporting their main claim, that CSR is regulated by DYRK1A through MSH6 phosphorylation.

We thank the reviewer for finding our study “original and well-presented”.

I have three main points to be addressed by the authors:

1. Experiments in Fig. 2E-G supporting a role for DYRK1A-mediated MSH6 phosphorylation promoting CSR should be further substantiated to establish this as the main mechanism by which DYRK1A impacts on CSR.

a. Can a MSH6 phosphomimic mutant bypass the CSR defects observed in DYRK1A deficient B-cells?

In an attempt to address this question we previously cloned and generated a lentivirus that expresses MSH6 with aspartic acid at position 326, and used this virus to transduce *Dyrk1a*-deficient B cells. Although we attempted this protocol twice, we did not get a sufficient number of GFP-positive cells in the B cell culture, whereas WT cells were effectively transduced. MSH6 is part of the DNA repair machinery, and we suspect that improper manipulation of this protein damages the KO cells either in a direct or indirect manner. We mentioned this problem in the text.

“Transduction of KO B cells with a virus carrying MSH6 T326D, which mimics the phosphorylated form, did not yield expressing cells, and as such, we could not measure the effect of constitutive phosphorylated MSH6.”

b. What is the effect of MSH6 phosphorylation on its mismatch repair activity? S-region mutation analyses in B-cells deficient for MSH6 show fewer A:T substitutions; a similar analysis in DYRK1A deficient B-cells or, ideally, B-cells expressing MSH6-T326A would help clarify how MSH6 phosphorylation impacts on mismatch repair during CSR.

We carefully examined the published literature regarding the role of MSH6 in nucleotide substitutions based on in vitro CSR assays. The most relevant paper is Li et al. 2004⁵¹. In this study, the authors found that only ~15% of the sequences of stimulated B cells were mutated, carrying only 1 or 2 mutations. Thus, the readout in this type of experiment is extremely weak.

Furthermore, the authors demonstrate that the G/C nucleotide substitutions were 66% of total mutations in WT cells and 76% in *Msh6*-deficient B cells. Thus, the effects of MSH6 deficiency are relatively mild and do not eliminate A/T nucleotide substitutions.

Despite these limitations, we invested major efforts in attempting to address the reviewer's request. We sequenced the upstream of the core S μ -region of B cells that were stimulated with LPS+IL4 for 3 days (20% of WT B cells were IgG1⁺) and analyzed a total of 361 S-regions from 2 mice per group. As expected, there were very few clones that carried mutations, as observed in Li et al. Nonetheless, although the number of mutations was small, we found more G/C substitutions in *Dyrk1a*-deficient B cells (supplementary figure 7).

These are very low numbers of mutations that cannot be examined by statistical tests. But given the very low efficiency of this assay, and the lack of strong effects of MSH6 on nucleotide substitutions⁵¹ in CSR, probably due to compensation mechanisms, it is unlikely that we will find a robust change in sequences recovered from *Dyrk1a*-deficient B cells.

We discuss these findings and their limitations in the main text.

“A similar analysis of SHM upstream of the core S μ -region done using LPS+IL-4 stimulated B cells for 3 days, revealed an increase in G/C nucleotide substitutions, as was previously shown in *Msh6*-deficient mice^{41,51}. However, since A/T substitutions were observed in *Msh6*-deficient mice and since in this assay, the mutation frequency is very low, the number of mutated sequences was not sufficient for statistical analysis (Supplementary Fig. 7).”

LPS+IL-4 3 days			
		CD23.Cre.Dyrk1a^{+/+}	CD23.Cre.Dyrk1a^{fl/fl}
		186 sequences	175 sequences
Substitution		22 mutations	9 mutations
		#	#
A to:	G	1	0
	T	0	0
	C	0	0
T to:	C	0	0
	A	3	0
	G	0	0
G to:	A	3	1
	T	4	0
	C	0	1
C to:	T	3	2
	A	7	3
	G	1	2

2. The analysis of nucleotide base substitutions in GC B-cells suggests that the main role for DYRK1A in the GC is not the regulation of SHM via MSH6. This raises questions as to why DYRK1A-mediated MSH6 phosphorylation is specifically needed for CSR. The authors

a. Is MSH6 phosphorylation by DYRK1A specifically required for MSH6 targeting at switch recombination sites?

b. Is MSH6 protein, through DYRK1A phosphorylation, involved in the generation and/or resolution of DSBs at switch recombination sites?

We thank the reviewer for noting these important points. We took advice from experts in the field, and we understand that this question is extremely difficult to address in the scope of the current study, as it requires deep molecular analyses and protein-DNA binding assays which do not fall in our current expertise. Of note, it remains to be clarified why AID targets the switch region in the Igh before the GC is seeded, and only the VDJ sequence during the GC reaction.

Nonetheless, we understand that this point is important, and we now discuss these open questions and the limitations of our study in the Discussion section. We do see some class-switched antibodies and B cells both in vitro and in vivo, suggesting that alternative or compensatory mechanisms occur in *Dyrk1a*-deficient B cells. We speculate that the chronic nature of the GC and continued activation of the DNA damage and repair mechanisms overcome the defect over time.

We now mention this point in the Discussion:

“The MutS α complex induces the insertion of additional SHMs into the immunoglobulin locus after the generation of nucleotide mismatch by AID activity^{41,50}. However, based on our findings, DYRK1A-mediated phosphorylation did not play a major role in this process. This suggests that a different overlapping molecular mechanism compensates for MSH6 functions in GC B cells, or that the chronic nature of the GC response allows accumulation of SHM through a different pathway⁵⁰. Whether MSH6 phosphorylation by DYRK1A is required for specific targeting of the IgH switch regions, and whether it regulates the generation or resolution of double-strand DNA breaks remains to be determined.”

3. It is difficult to reconcile that in AID-Cre DYRK1A fl/fl mice, an enlarged GC phenotype is concomitant with a diminished dark-zone compartment, where most GC B-cell proliferation occurs.

a. The authors suggests that an increase in GC size in AID-Cre DYRK1A fl/fl mice is the result of enhanced proliferation in the light-zone. This should be directly tested using dual-labelling with EdU and BrdU, to determine cell-cycle dynamics in both dark-zone and light-zone compartments.

We examined B cell proliferation and S-phase entry using EdU and BrdU labeling, as suggested by the reviewer. We found enhanced B cell proliferation in the DZ but not in the LZ (Supplementary Fig. 8). We would like to clarify that the DZ is not substantially diminished, but rather is slightly smaller compared to the control mice, yet the GC size is significantly increased in *AID.Cre.Dyrk1a^{fl/fl}* mice. Indeed, we found a 4-fold increase in the number of antigen-specific B cells in the GCs of the *AID.Cre.Dyrk1a^{fl/fl}* mice compared to control. Together, these results suggest that the increased GC size in *AID.Cre.Dyrk1a^{fl/fl}* mice is most likely due to enhanced proliferation that occurs in the DZ.

We now modified our initial prediction in the text.

“Enhanced proliferation, but not initiation of cell cycle, was detected in DZ B cells and not in LZ B cells following dual-labeling with EdU and BrdU.”

“The rate of B cell proliferation in the DZ of *AID.Cre.Dyrk1a^{fl/fl}* mice was higher compared to control animals, suggesting that a defect in the pathways restraining the cell cycle in this zone by DYRK1A can explain the increase in global GC size.”

Supplementary Fig. 8

b. DYRK1A has been shown to control pre-B cell proliferation through destabilization of CyclinD3 protein levels upon phosphorylation. CyclinD3 drives 'inertial' cell cycle in dark-zone B-cells through sustained E2F expression – do the authors find evidence for changes in CyclinD3 protein levels in DYRK1A deficient GC B-cells?

Using flow cytometry, we were able to detect the different levels of CyclinD3 in different cell types. Yet, we did not observe more CyclinD3 protein in *Dyrk1a*-deficient B cells (Supplementary Fig. 1).

We therefore added the following sentences to the text:

“It was previously shown that DYRK1A phosphorylates c-Myc in acute myeloid leukemia (AML), and Cyclin D3 in pre-B cells to enhance their degradation^{28,33}. Using flow cytometry, we did not detect measurable changes in the expression of these cell-cycle regulators in either naive, GC B cells, or plasma cells (Supplementary Fig. 1).”

“Although cell proliferation pathways were affected and enhanced in *Dyrk1a*-deficient B cells, no defect in c-Myc and Cyclin D3 protein levels was detected.”

Supplementary Fig. 1

REVIEWERS' COMMENTS

Reviewer #1 (Remarks to the Author):

The authors successfully addressed my (major) points by performing experiments and making textual alterations. These new experiments make the combined dataset more convincing and in line with the main conclusions.

There is however one remaining minor point that needs adjustment:

It is important to formulate the role of BCL-2 family proteins in the GC correctly and in line with the literature indicated by the authors. BCL-2 itself is not expressed in the GC, but BCL-2 family members BCL-XL and MCL-1 are. Therefore please correct the newly introduced sentence "BCL-2 reduces.....GC size" into "BCL-2 family proteins BCL-XL and MCL-1 reduce....GC size".

Reviewer #2 (Remarks to the Author):

The authors have addressed all of my concerns and I am happy to accept for publication.

Reviewer #3 (Remarks to the Author):

The authors have carefully addressed the main points raised, and changed the text accordingly. I agree that although it would be important to understand the mechanisms by which MSH6 phosphorylation by DYRK1 is specifically required for CSR, this question would require a significant amount of work to be addressed fully. I appreciate the authors have now acknowledge this point as an open question in the Discussion.

Before publication, I would like to request the authors clarify the following conclusion:

"Enhanced proliferation, but not initiation of cell cycle, was detected in DZ B cells and not in LZ B cells following dual-labeling with EdU and BrdU."

From the results shown in Supplementary Fig.8a (i.e. increase in early S, EdU neg BrdU pos DZ cells) I would conclude that initiation of cell cycle is increased in Dyrk1-deficient DZ B-cells.

Point by point response to the reviewers

Reviewer #1 (Remarks to the Author):

The authors successfully addressed my (major) points by performing experiments and making textual alterations. These new experiments make the combined dataset more convincing and in line with the main conclusions.

There is however one remaining minor point that needs adjustment:

It is important to formulate the role of BCL-2 family proteins in the GC correctly and in line with the literature indicated by the authors. BCL-2 itself is not expressed in the GC, but BCL-2 family members BCL-XL and MCL-1 are. Therefore please correct the newly introduced sentence “BCL-2 reduces.....GC size” into “BCL-2 family proteins BCL-XL and MCL-1 reduce....GC size”.

“BCL-2 family proteins BCL-XL and MCL-1 reduce B cell apoptosis in the GC, and its enhanced expression is associated with increased GC size” was added to the text.

Reviewer #2 (Remarks to the Author):

The authors have addressed all of my concerns and I am happy to accept for publication.

Reviewer #3 (Remarks to the Author):

The authors have carefully addressed the main points raised, and changed the text accordingly. I agree that although it would be important to understand the mechanisms by which MSH6 phosphorylation by DYRK1 is specifically required for CSR, this question would require a significant amount of work to be addressed fully. I appreciate the authors have now acknowledge this point as an open question in the Discussion.

Before publication, I would like to request the authors clarify the following conclusion: “Enhanced proliferation, but not initiation of cell cycle, was detected in DZ B cells and not in LZ B cells following dual-labeling with EdU and BrdU.”

From the results shown in Supplementary Fig.8a (i.e. increase in early S, EdU neg BrdU pos DZ cells) I would conclude that initiation of cell cycle is increased in Dyrk1-deficient DZ B-cells.

“Enhanced proliferation and initiation of cell cycle, were detected in DZ B cells and not in LZ B cells following dual-labeling with EdU and BrdU” was added to the text.